# ASH1L histone methyltransferase regulates the handoff between damage recognition factors in global-genome nucleotide excision repair

Chiara Balbo Pogliano [1], Marco Gatti[2,3], Peter Rüthemann[1], Zuzana Garajovà[1], Lorenza Penengo[3] & Hanspeter Naegeli [1]

Global-genome nucleotide excision repair (GG-NER) prevents ultraviolet (UV) light-induced skin cancer by removing mutagenic cyclobutane pyrimidine dimers (CPDs). These lesions are formed abundantly on DNA wrapped around histone octamers in nucleosomes, but a specialized damage sensor known as DDB2 ensures that they are accessed by the XPC initiator of GG-NER activity. We report that DDB2 promotes CPD excision by recruiting the histone methyltransferase ASH1L, which methylates lysine 4 of histone H3. In turn, methylated H3 facilitates the docking of the XPC complex to nucleosomal histone octamers. Consequently, DDB2, ASH1L and XPC proteins co-localize transiently on histone H3-methylated nucleosomes of UV-exposed cells. In the absence of ASH1L, the chromatin binding of XPC is impaired and its ability to recruit downstream GG-NER effectors diminished. Also, ASH1L depletion suppresses CPD excision and confers UV hypersensitivity. These findings show that ASH1L configures chromatin for the effective handoff between damage recognition factors during GG-NER activity.

[1] Institute of Pharmacology and Toxicology, University of Zurich-Vetsuisse, Winterthurerstrasse 260, 8057 Zurich, Switzerland. [2] Department of Molecular Mechanisms of Disease, University of Zurich, Winterthurerstrasse 190, 8057 Zurich, Switzerland. [3] Institute of Molecular Cancer Research, University of Zurich, Winterthurerstrasse 190, 8057 Zurich, Switzerland. Correspondence and requests for materials should be addressed to H.N. (email: naegelih@vetpharm.uzh.ch)

Genomic DNA is attacked by multiple genotoxic insults. In particular, the ultraviolet (UV) radiation of sunlight induces crosslinks between neighboring bases to generate mainly cyclobutane pyrimidine dimers (CPDs)[1, 2]. These highly mutagenic CPD lesions are induced evenly in chromatin and arise abundantly in nucleosome cores where the DNA is wrapped around histone octamers[3, 4]. The versatile nucleotide excision repair (NER) system removes UV lesions and other bulky base adducts generated by chemical carcinogens or oxygen radicals[5–7]. Depending on their location in the genome, base lesions are sensed by two alternative pathways. In transcription-coupled NER (TC-NER), damage detection occurs when RNA polymerase II runs into obstructing adducts in the template strand[8, 9]. Instead, the vast majority of DNA adducts are recognized by global-genome NER (GG-NER) independently of transcription[10, 11]. The importance of this global pathway is demonstrated by the extreme solar hypersensitivity and skin cancer incidence of xeroderma pigmentosum (XP) patients[12, 13].

Subjects afflicted by this hereditary disease are classified into complementation groups (XP-A through XP-G) carrying mutations in different NER genes[14, 15].

The GG-NER reaction uses a trimeric factor comprising XPC, RAD23B (a human homolog of yeast RAD23) and centrin 2 to sense DNA lesions[16–19]. XPC is the subunit that binds to DNA and, for the recognition of CPDs, this repair initiator is assisted by an auxiliary factor with damaged DNA-binding (DDB) activity[20–24]. DDB2 is the actual UV damage sensor, which through the DDB1 adapter associates with the cullin 4 A (CUL4A) ubiquitin ligase[25–27]. By a yet unclear mechanism, DDB2 hands off UV lesions to the XPC subunit, which in turn recruits transcription factor IIH (TFIIH) containing the XPD helicase whose function is to unwind and scan DNA for damage verification[28–30]. The resulting intermediate is stabilized by XPA and replication protein A (RPA)[31] until endonucleases (XPG and a heterodimer of XPF and excision repair cross-complementing 1) incise the damaged strand on either side of the unwound helix.

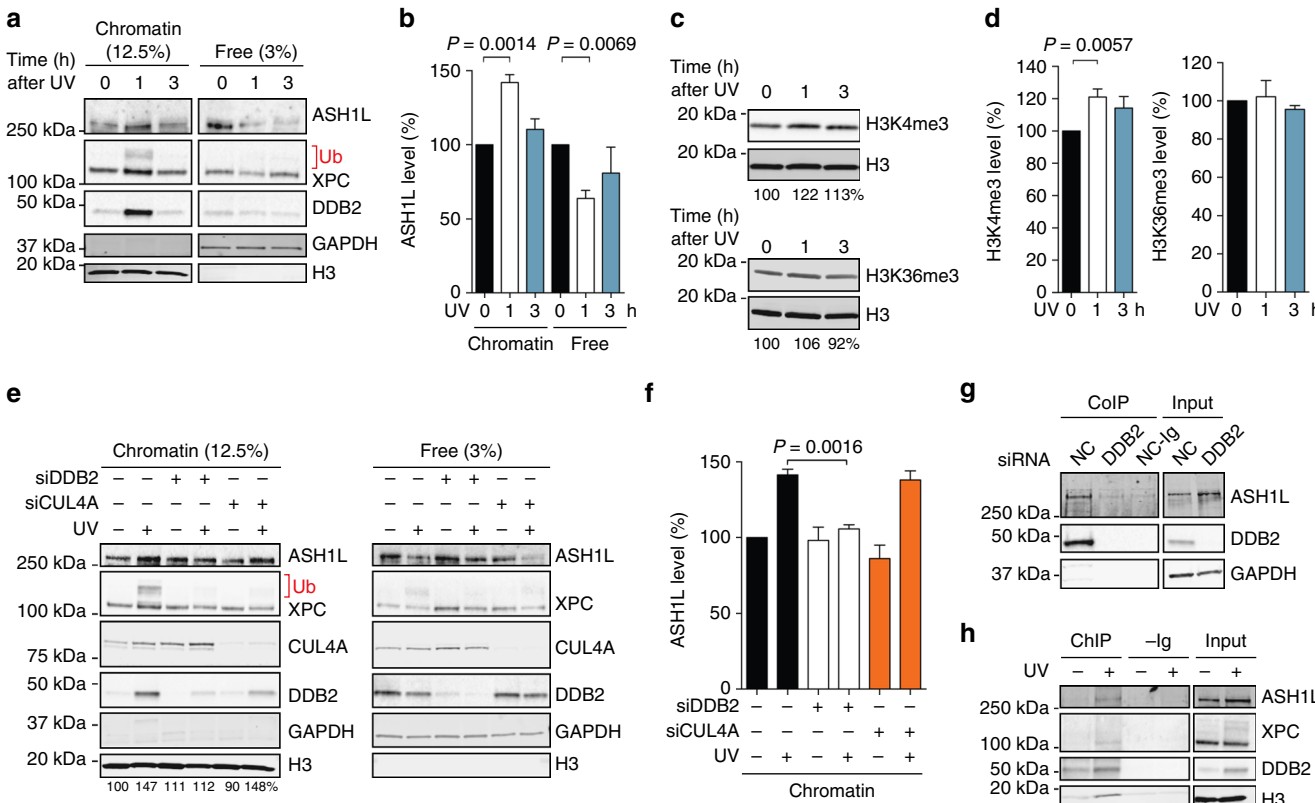

**Fig. 1** DDB2-dependent recruitment of ASH1L to chromatin. **a** Recruitment of DDB2, XPC and ASH1L to chromatin detected by immunoblotting. The chromatin of HeLa cells exposed to UV-C (10 J m$^{-2}$) was analyzed, at different times, by salt extraction to release free proteins, followed by MNase solubilization. GAPDH and H3 were loading controls for free proteins and solubilized chromatin, respectively. Ub, ubiquitinated XPC protein. **b** Quantification of ASH1L recruitment to chromatin normalized to H3 ($n = 4$ independent experiments). ASH1L levels in unirradiated chromatin are set to 100%; error bars show s.e.m. and significance was calculated using the one-sample t-test. **c** Increased H3K4me3 levels following UV radiation (10 J m$^{-2}$). Whole cell lysates were probed with antibodies against the indicated forms of H3. Numbers at the bottom of each blot indicate the quantity of methylated H3 normalized to total H3 and relative to unirradiated controls. **d** Quantification of H3K4me3 and H3K36me3 normalized to total H3 ($n = 6$ independent experiments, one-sample t-test), values in unirradiated controls are set to 100%. **e** DDB2-dependent recruitment of ASH1L to chromatin. HeLa cells were siRNA-transfected as indicated 48 h before UV radiation (10 J m$^{-2}$). After a 1-h recovery, cells were analyzed by salt extraction, chromatin solubilization and immunoblotting. NC, non-coding. Numbers at the bottom of the blot indicate ASH1L protein levels normalized to H3 and relative to the unirradiated control. **f** Quantification of ASH1L recruitment to chromatin, relative to free ASH1L, normalized to H3 and GAPDH ($n = 4$ independent experiments, one-sample t-test). **g** Co-immunoprecipitation (Co-IP) of ASH1L with DDB2. HeLa cells were siRNA-transfected 48 h before collection of 0.3-M NaCl extracts. These extracts were immunoprecipitated with anti-DDB2 antibodies and analyzed by immunoblotting; –Ig, immunoglobulins omitted (the DDB2-proficient extract was subjected to the same Co-IP procedure except that anti-DDB2 antibodies were left out). **h** Chromatin immunoprecipitation (ChIP) using antibodies against DDB2 demonstrating that DDB2 and ASH1L interact in the chromatin of UV-irradiated (10 J m$^{-2}$) HeLa cells; –Ig, immunoglobulins omitted (samples were subjected to the same ChIP procedure except that anti-DDB2 antibodies were left out)

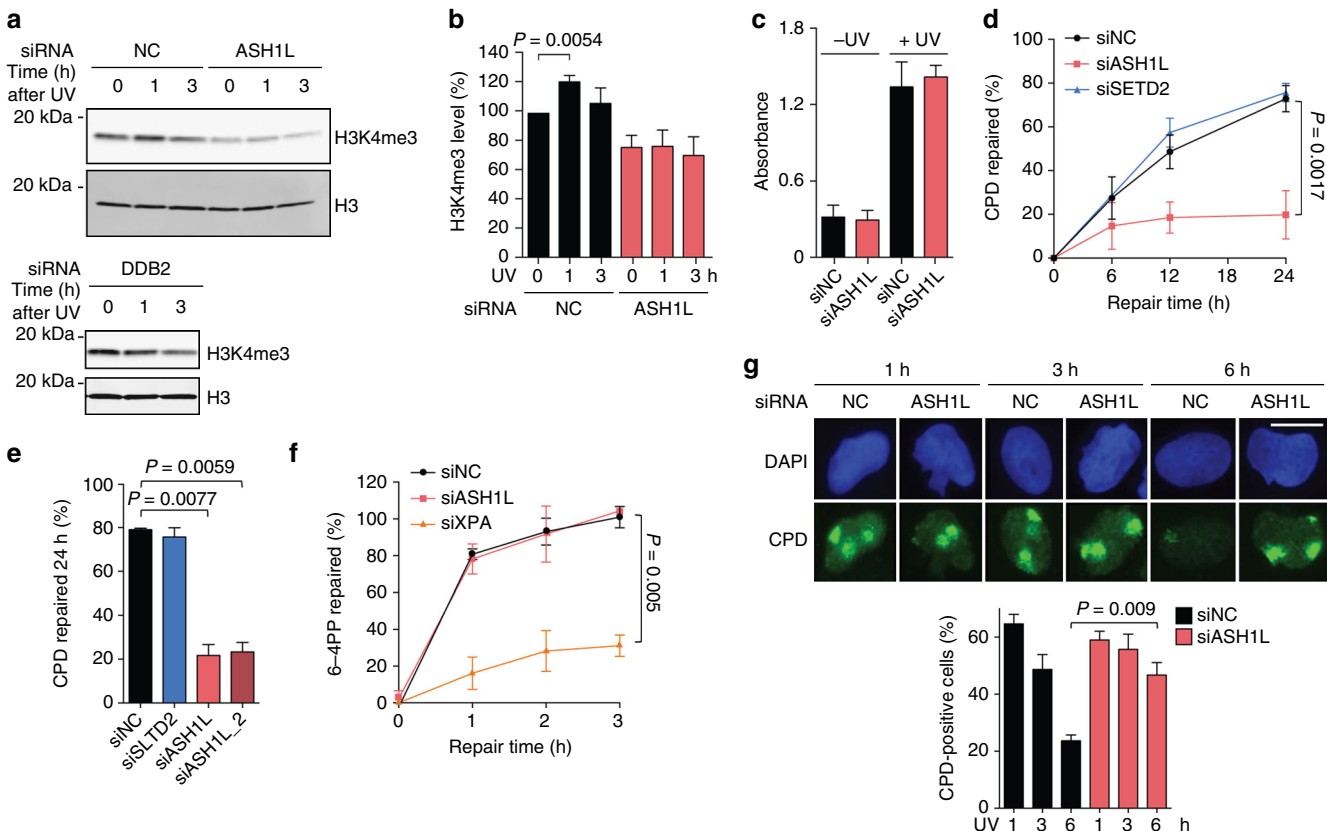

**Fig. 2** Impaired CPD excision upon ASH1L depletion. **a** Visualization of H3K4me3 in ASH1L-depleted HeLa cells. Whole cell lysates were analyzed by immunoblotting using antibodies against the indicated forms of H3. UV radiation was at 10 J m$^{-2}$. **b** Quantification of H3K4me3 levels normalized to total H3 ($n = 6$ independent experiments). Error bars show s.e.m. (one-sample t-test). **c** Initial CPD formation after UV irradiation. HeLa cells were transfected with siASH1L or siNC, UV-irradiated (10 J m$^{-2}$) 48 h later and immediately collected for CPD analysis. Immunoassay absorbance, providing a measure of CPDs, were not affected by the ASH1L depletion ($n = 3$, each experiment with 4 replicates; two-tailed t-test). **d** Excision of CPDs in HeLa cells treated 48 h before UV radiation (10 J m$^{-2}$) with siRNA targeting ASH1L or SETD2, compared to transfections with siNC ($n = 5$, each experiment with 4 replicates, two-tailed t-test). **e** Excision of CPDs in HeLa cells transfected 48 h before UV radiation (10 J m$^{-2}$) with siASH1L, siASH1L_2 or siSETD2, compared to siNC. CPDs were measured immediately after UV treatment and after a 24-h repair period ($n = 3$ with 4 replicates each, two-tailed t-test). **f** Excision of 6–4PPs in HeLa cells transfected 48 before UV radiation (10 J m$^{-2}$) with siRNA targeting ASH1L in comparison to siXPA and siNC ($n = 3$ with 4 replicates each, two-tailed t-test). **g** Immunofluorescence detection of CPDs in cells transfected with siASH1L or siNC, 48 h before UV radiation (100 J m$^{-2}$) through 5-μm filter pores. DNA was stained with DAPI. The quantification of nuclei positive for CPD spots is based on a minimum of 100 cells ($n = 3$ independent experiments, two-tailed t-test). Scale bar = 15 μm. Representative wide-field views of cells are shown in Supplementary Fig. 6

Damaged bases are removed as part of an oligonucleotide of 24–32 residues[32, 33] and the excision gap is processed by DNA repair synthesis and ligation[34, 35].

How GG-NER activity takes place despite DNA packaging in nucleosomes is currently under intense scrutiny. Nucleosomes are the building block of chromatin and consist of core particles separated by linker DNA of variable length. In each nucleosome core, 147 base pairs of DNA are wrapped around a histone octamer, i.e., two copies each of H2A, H2B, H3 and H4. These core histones present *N*-terminal tails undergoing modifications that regulate chromatin dynamics. For example, lysine acetylation in histone tails facilitates accessibility of the GG-NER complex to damaged DNA[36–40]. Histone acetylation neutralizes positive charges and, therefore, weakens directly DNA-histone interactions to favor DNA transactions by relaxing nucleosomes[41]. A different mechanism occurs upon histone methylation as the resulting methylated tails provide hydrophobic docking sites for factors inducing chromatin condensation or relaxation[42]. In general, methylation of lysines K4 and K36 of histone H3 correlates with relaxed chromatin[43–45].

The present study was prompted by the finding that the histone methyltransferase ASH1L (for *A*bsent, *S*mall, or *H*omeotic discs *1-L*ike), a member of *trithorax* transcriptional regulators essential for development, organ function and fertility[46, 47], can associate with chromatin independently of ongoing transcription[48]. This observation raised the possibility that ASH1L may exert pleiotropic roles in regulating chromatin states for various DNA functions. Indeed, we identify this particular histone methyltransferase as an accessory player coordinating the substrate handover from DDB2 to XPC during initiation of the GG-NER reaction in the nucleosome landscape. We demonstrate that ASH1L is recruited to chromatin by the lesion sensor DDB2. Upon UV irradiation, ASH1L generates lysine 4-trimethylated histone H3K4me3, which promotes the stable docking of XPC protein to nucleosomes. An XPC mutation that disrupts this ASH1L-dependent interaction with core histones results in defective CPD repair. Thus, ASH1L regulates the handoff between DDB2 and XPC required to initiate GG-NER activity.

## Results

**UV-dependent ASH1L recruitment and histone methylation.** At least one histone methyltransferase known as SETD2 has been shown to participate in DNA mismatch repair[49] and

recombination[50–52]. To test their involvement in the UV radiation response, we transfected HeLa cells with a range of siRNA sequences targeting SETD2 and further histone methyltransferases. This siRNA screen suggested that several of these enzymes contribute to survival after UV exposure. In a comparison of cell viability 48 h after UV irradiation, ASH1L down regulation conferred a stronger UV hypersensitivity than depletion of other histone methyltransferases (Supplementary Fig. 1). Based on this initial screen, we explored the role of ASH1L in the processing of UV lesions.

We first tested whether ASH1L translocates to the chromatin of human cells following UV radiation. HeLa cells were lysed and extracted with 0.3 M NaCl to remove, into the supernatant, free proteins that are not associated with chromatin or only loosely bound to chromatin. The remaining chromatin pellet was solubilized by digestion with micrococcal nuclease (Mnase) that cleaves DNA in linker segments between nucleosome cores (see flow diagram in Supplementary Fig. 2). This MNase-solubilized chromatin was inspected by immunoblotting to monitor protein recruitment to damaged DNA. As expected, UV treatment of the cells resulted in increased levels of DDB2 and XPC, the two initiators of GG-NER activity, in chromatin (Fig. 1a). In line with previous reports[27, 53] part of chromatin-associated XPC is ubiquitinated after UV irradiation. Also, the UV treatment led to an additional recruitment of ASH1L over the constitutive presence of this methyltransferase in chromatin of unchallenged cells, accompanied by a concomitant decrease of free ASH1L in the 0.3-M NaCl supernatant. The immunoblot quantifications using histone H3 and glyceraldehyde 2-phosphate dehydrogenase (GAPDH) as standards showed that the level of ASH1L in chromatin is increased by ~ 40% around 1 h after UV irradiation compared to unirradiated controls (Fig. 1b). ASH1L methylates histone H3 at the positions K4 and K36[46, 48]. Therefore, we tested whether the UV-dependent recruitment of ASH1L involves changes in H3 methylation. Using anti-H3K4me3 antibodies, we observed 1 h after UV radiation a consistent increase in the level of methylated histone H3 (Fig. 1c, d). No such increase was observed for H3K36me3 at any time after UV exposure.

**DDB2 mediates the UV-dependent recruitment of ASH1L**. To test the mechanism by which ASH1L is recruited to chromatin in response to UV light, HeLa cells were depleted of the DDB2 lesion recognition subunit by transfection with siRNA (see Supplementary Fig. 3 for the efficiency of down regulation), followed by UV irradiation and chromatin analysis. As already seen in Fig. 1a, UV treatments resulted in an increase of ASH1L bound to chromatin, but this additional recruitment over the constitutive level was abrogated by DDB2 down regulation (Fig. 1e, f). Depletion of the CUL4A ubiquitin ligase scaffold affected partially the amount of DDB2 bound to chromatin, but without interfering with the extra UV-dependent chromatin localization of ASH1L. These findings indicate that in the absence of CUL4A ubiquitin ligase activity lower amounts of DDB2 are sufficient for the recruitment of ASH1L and, in any case, imply that the DDB2 subunit itself, rather than the associated ubiquitin ligase complex, is required for the UV-dependent ASH1L redistribution to chromatin.

Next, we exploited the 0.3-M NaCl supernatant of unchallenged cells, containing free DDB2 and ASH1L (Fig. 1a), to test potential interactions between these two factors. Immunoprecipitation of DDB2 from cell extracts resulted in the co-precipitation of ASH1L protein. However, the methyltransferase was missing in the immunoprecipitated fraction when control reactions were carried out with DDB2-depleted cells (Fig. 1g). We then tested whether this interaction with DDB2 may mediate the

UV-dependent ASH1L recruitment. In a chromatin-immunoprecipitation assay, antibodies against DDB2 were able to co-isolate ASH1L preferentially from UV-irradiated cells (Fig. 1h). Thus, an interaction occurs between these two factors such that DDB2 is able to mediate the relocation of ASH1L to UV lesions in the chromatin context.

**ASH1L depletion suppresses CPD excision**. HeLa cells were depleted of ASH1L by transfection with siRNA as shown in Supplementary Fig. 3. The lack of ASH1L affected only partially the constitutive H3K4me3 content. In the absence of ASH1L, however, the UV-dependent increase of H3K4me3 observed 1 h after UV irradiation is abrogated (Fig. 2a, b). This increase of H3K4me3 observed 1 h after UV irradiation is also missing in the absence of DDB2 (Fig. 2a). Considering its function in generating H3K4me3 following UV damage and this dependency on DDB2, we tested whether ASH1L may have a role in the GG-NER process. HeLa cells were UV-irradiated and the presence of lesions in their genome was monitored by immunoassays. We established that a preceding ASH1L depletion does not change the amount of CPDs detected immediately after the UV pulse (Fig. 2c), indicating that the presence or absence of ASH1L does not change the susceptibility for lesion formation. However, CPD excision was severely reduced under conditions of ASH1L depletion. After 24 h of repair, ~ 70% of CPDs were removed in cells transfected with control RNA or a sequence against SETD2. Instead, CPD excision during this 24-h period was limited to only ~ 20% in ASH1L-depleted cells (Fig. 2d). This marked effect with only ~ 20% residual excision in 24 h was confirmed using a distinct siRNA sequence targeting ASH1L (Fig. 2e). As GG-NER contributes by 90–95% to the overall excision of DNA adducts and TC-NER only by the remaining 5–10%[54], the drop in excision rate observed upon ASH1L depletion is consistent with an involvement of this histone methyltransferase in the GG-NER reaction. Conversely, the ASH1L depletion has no consequences on the excision of (6–4) photoproducts (6–4PPs; Fig. 2f), which are formed three times less frequently than CPDs and arise mainly in linker segments[1, 55].

We exploited an independent method based on *in situ* immunofluorescence to visualize the different CPD excision rate upon ASH1L depletion compared to controls. In view of their large nucleus, U2OS cells provide a more amenable substrate for such immunofluorescence studies than HeLa cells. Therefore, U2OS cells were UV-irradiated through 5-μm filter pores to generate nuclear spots of damage containing CPDs. Following incubations of 1, 3 or 6 h, cells were fixed, permeabilized and stained with anti-CPD antibodies. Fluorescence quantifications demonstrated that detectable CPD spots disappear with time in control cells transfected with non-coding RNA. As a consequence of the lower signal-to-noise ratio of this in situ method compared to the immunoassay of Fig. 2d, remaining CPDs in UV damage spots were barely visible over the nuclear background following 6 h of repair. Instead, in ASH1L-depleted cells the fluorescence intensity due to CPDs in UV damage spots remained at 6 h nearly as after 1 h of incubation (Fig. 2g). Taken together, these results show that ASH1L is required for efficient CPD excision.

**ASH1L depletion impedes GG-NER and confers UV sensitivity**. The NER pathway excises DNA adducts as part of oligonucleotides of 24–32 residues, which are replaced by DNA repair synthesis. Accordingly, we tested whether the reduced CPD excision in ASH1L-depleted cells translates to lower repair patch synthesis. Spots of UV damage were generated in the nuclei of U2OS cells and, to measure DNA synthesis, these cells were supplemented with the nucleoside analog 5-ethynyl-2′-

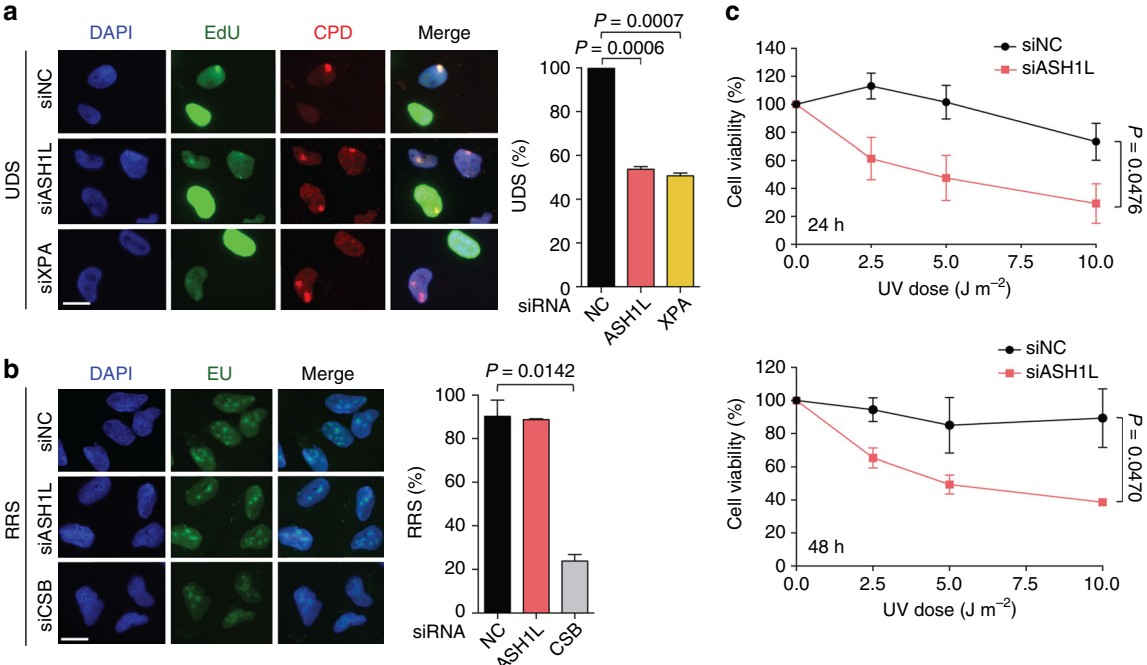

**Fig. 3** Impaired GG-NER activity upon ASH1L depletion. **a** Unscheduled DNA synthesis (UDS). U2OS cells were transfected with siRNA and UV-irradiated (100 J m$^{-2}$) through 5-μm filter pores 48 h later. EdU was added after a 2-h recovery to allow for 6–4PP excision, and fluorescence reflecting UDS was measured after another 1-h period. S-phase cells were excluded from quantification ($n = 3$ with at least 100 cells per experiment). The UDS detected in control cells transfected with non-coding (NC) RNA is set to 100% and error bars show s.e.m. (one-sample t-test); scale bar = 15 μm. **b** Recovery of RNA synthesis (RRS). U2OS cells were transfected with siRNA and globally UV-irradiated (10 J m$^{-2}$) 48 h later. Next, the cells were allowed to recover for 4 h before the addition of EU for 2 h. The resulting nuclear fluorescence, shown as the percentage of the corresponding values in unirradiated controls, reflects RNA synthesis ($n = 3$ with at least 100 cells per experiment; two-tailed t-test). **c** UV light sensitivity. HeLa cells were irradiated with the indicated UV doses and their viability was measured after 24 and 48 h ($n = 3$, each experiment with 4 replicates; two-tailed t-test)

deoxyuridine (EdU) following a 2-h recovery after irradiation, i.e., after removal of most 6–4PPs as indicated in Supplementary Fig. 4. The fluorescence intensity in CPD spots resulting from EdU incorporation during 1 h, demonstrated that, like the XPA depletion, an ASH1L deficiency results in much lower levels of repair synthesis compared to controls (Fig. 3a). This finding confirms that ASH1L is required for efficient processing of CPDs by the GG-NER reaction.

Conversely, TC-NER can be monitored by the recovery of transcription after DNA damage. UV radiation causes a decrease of RNA synthesis, which recovers readily in normal cells due to TC-NER activity[54, 56]. We globally irradiated U2OS cells, allowed them to recover for 4 h, added 5-ethynyl uridine (EU) for another 2 h and, after this overall 6-h incubation, measured EU-linked fluorescence reflecting RNA synthesis across cell nuclei. Importantly, the recovery of RNA synthesis was heavily delayed in cells depleted of Cockayne syndrome group B (CSB) protein required for the TC-NER reaction. In contrast, the ASH1L depletion did not interfere at all with RNA synthesis recovery (Fig. 3b) indicating that, although this histone methyltransferase regulates GG-NER activity, it is not involved in the TC-NER reaction. The role of ASH1L in the processing of CPDs is confirmed by the observation that its down regulation confers hypersensitivity to the cytotoxic effect of UV light (Fig. 3c).

**ASH1L depletion dysregulates GG-NER assembly**. In view of the finding that ASH1L stimulates GG-NER but not TC-NER activity, we tested whether this histone methyltransferase may affect the expression of XPC and DDB2, which are the only NER factors solely required for the global-genome subpathway.

Analysis of HeLa whole cell lysates showed, however, that the expression of XPC is rather increased by ASH1L depletion (Fig. 4a, b). After UV irradiation, part of XPC protein appears in a high-molecular weight form resulting from ubiquitination[27] but its overall cellular level remains increased in ASH1L-depleted cells compared to ASH1L-proficient controls. Conversely, the ASH1L depletion did not change the constitutive DDB2 level in unchallenged cells. As expected[23], DDB2 is degraded upon UV radiation and we noted that this UV-dependent DDB2 decline is slower in ASH1L-depleted cells (Fig. 4a, b). These findings allow us to conclude that the GG-NER defect observed in ASH1L-depleted cells cannot be attributed to reduced levels of the two factors (XPC and DDB2) specifically involved in this pathway. Notably, the cellular content of XPD, involved in both GG-NER and TC-NER, remained unaffected by the ASH1L depletion (Fig. 4a).

We then carried out fluorescence microscopy to test whether ASH1L regulates the recruitment of XPC protein to chromatin. U2OS cells were transfected with siRNA against ASH1L (or with non-coding RNA) and, 48 h later, UV-irradiated through 5-μm filter pores to generate UV lesion spots. After different times, cells were fixed and permeabilized to monitor the *in situ* NER factor distribution by immunofluorescence[57]. At 1 h after radiation, the XPC accumulation in UV lesion spots of ASH1L-depleted cells is not different from that in cells transfected with non-coding RNA (Fig. 5a). In these controls, the XPC occupancy of UV lesion sites decreases with time such that, 3 h after radiation, only few cells display discernible XPC spots co-localizing with CPDs. Upon ASH1L depletion, instead, XPC is retained at UV lesions and, 3 h after radiation, the majority of cells still display bright XPC spots co-localizing with CPDs. A marked difference between control

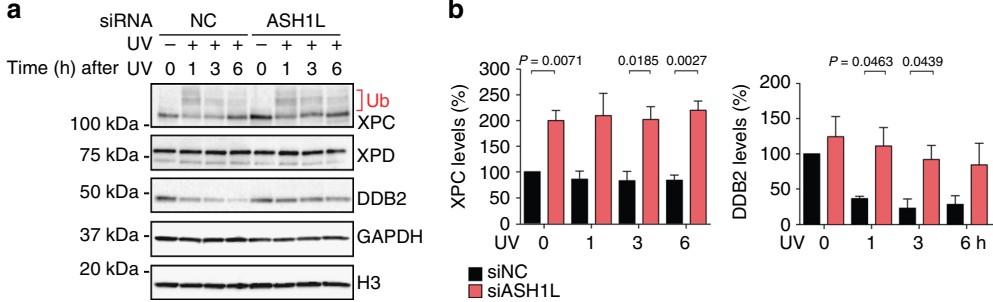

**Fig. 4** Increased XPC expression upon ASH1L depletion. **a** Immunoblot analysis of XPC, XPD and DDB2 in whole cell lysates. HeLa cells were pre-treated with siRNA targeting ASH1L or non-coding control RNA (NC), UV-irradiated (10 J m$^{-2}$) 48 h later and collected at different times. GAPDH and H3 were used as loading controls. **b** Level of XPC and DDB2 normalized to H3 ($n = 4$ independent experiments). Quantifications include both unmodified and ubiquitinated XPC. Protein levels in unirradiated controls are set to 100% and error bars show s.e.m. (one-sample t-test)

and ASH1L-depleted cells was also detected 3 h after radiation by measuring fluorescence intensity representing the level of XPC in each individual spot. For these quantifications, the fluorescence at damaged spots was divided by the background nuclear signal. This procedure ensures that the data demonstrate a truly increased XPC retention at CPD sites rather than reflecting its overall higher presence following ASH1L depletion. A simultaneous down regulation of both ASH1L and DDB2 demonstrates that this prolonged retention of XPC at UV spots, observed 3 h after irradiation, disappears in the absence of DDB2 protein (Fig. 5a). Surprisingly, the prolonged XPC retention at damaged sites of ASH1L-depleted but DDB2-proficient cells translates to a reduced recruitment of the downstream XPD helicase, which as part of the TFIIH complex is responsible for DNA damage verification. An impaired XPD relocation to UV lesions is detected 3 h after UV radiation both by monitoring the percentage of cells with XPD spots co-localizing with CPDs, and by quantifying the XPD level at each individual spot relative to the nuclear background (Fig. 5b). We conclude that ASH1L is required to position XPC protein onto chromatin in a way that this repair initiator can form a productive association with CPD sites leading to recruitment of the XPD helicase.

**ASH1L ensures the stable association of XPC with CPD sites.** To understand how ASH1L regulates XPC protein, U2OS cells were again transfected with siRNA against ASH1L (or with non-coding RNA) and, 48 h later, UV-irradiated through filters with 3-μm pores to generate "small" spots of UV lesions. After 2-h incubations to allow for repair of 6–4PPs, cells were irradiated through 5-μm pores to induce "big" spots of UV lesions. After another incubation of 15 min, the cells were fixed and permeabilized to monitor the *in situ* relocation of XPC protein by immunofluorescence (Fig. 6a). Control cells transfected with non-coding RNA displayed XPC accumulations in both small and big spots (Fig. 6b), reflecting the ability to engage with not yet processed CPDs in small spots as well as with the newly formed UV lesions, including 6–4PPs, in big spots. Upon ASH1L depletion, however, XPC is almost completely relocated to big spots (Fig. 6b), indicating that in the absence of ASH1L it is not able to stably interact with CPD sites and that, as a consequence, XPC protein readily moves to 6–4PPs in the newly formed big spots.

The results obtained so far indicate that DDB2 attracts XPC protein to the vicinity of CPDs and that ASH1L is then needed for XPC protein to stably interact with CPD sites. To test this scenario, HeLa cells were transfected with siRNA targeting ASH1L (or with non-coding RNA) and, after 48 h, exposed to UV light or mock-treated. Cells were collected 1 and 3 h after treatment and, without prior fixation, subjected to 0.3-M NaCl

extraction to remove free proteins that are not or only loosely bound to chromatin. The remaining chromatin was analyzed by gel electrophoresis and immunoblotting, and quantifications took into account the generally higher level of XPC protein in ASH1L-depleted cells. As reported before[58], a substantial proportion of the XPC subunit is constitutively bound to chromatin both in control cells and ASH1L-depleted cells even without radiation. In control cells, the level of XPC protein recruited to chromatin is increased 1 h after UV exposure relative to the unchallenged situation (Fig. 6c, d). However, in ASH1L-depleted cells, this extra UV-dependent XPC recruitment to chromatin compared to the respective unirradiated reference is much lower, resulting in higher proportions of XPC in the fraction of free proteins. In contrast, the time-dependent changes of chromatin-bound DDB2 levels in ASH1L-depleted cells are indistinguishable from those in controls, implying that ASHIL does not influence the assembly of DDB2 with UV lesions and its dynamic turnover at sites of damage. Conversely, the missing XPC relocation to chromatin in ASH1L-depleted cells exposed to UV radiation also causes an impaired recruitment of the XPD helicase (Fig. 6c, d). Therefore, this biochemical analysis of Fig. 6c and the *in situ* dynamics of Fig. 6b converge on the finding that ASH1L is required to position XPC protein onto chromatin such that this repair initiator can form stable associations with CPD sites allowing for the recruitment of the XPD helicase.

**XPC binds preferentially to nucleosomes containing H3K4me3.** To clarify the mechanism by which ASH1L supports the XPC relocation, we examined the nucleosomes of HeLa cells that were either UV-irradiated or mock treated. Following a 1-h recovery, the chromatin of these cells was extracted with 0.3 M NaCl (to remove unbound or loosely bound proteins) and core particles were released by MNase digestion (see Supplementary Fig. 2). Analysis of DNA lengths demonstrates the complete digestion of linker segments leaving monomeric core fragments of 147 base pairs (Supplementary Fig. 5). Analysis of the protein composition of this solubilized fraction confirmed the increased chromatin localization of DDB2, XPC and ASH1L upon UV radiation (Fig. 7a). Immunoprecipitation with antibodies against methylated H3K4me3 or against total H3 shows that a substantial part of XPC protein localizing to chromatin is associated with the histones of nucleosome core particles. In addition, this side-by-side comparison demonstrated that XPC protein associates preferentially with core particles containing methylated H3K4me3 (precipitated with anti-H3K4me3 antibodies) as compared to those containing mainly unmethylated H3 (precipitated with anti-H3 antibodies). ASH1L is preferentially bound to its own product, i.e., nucleosomes containing modified H3K4me3. Instead, DDB2 preferentially associates with nucleosome core

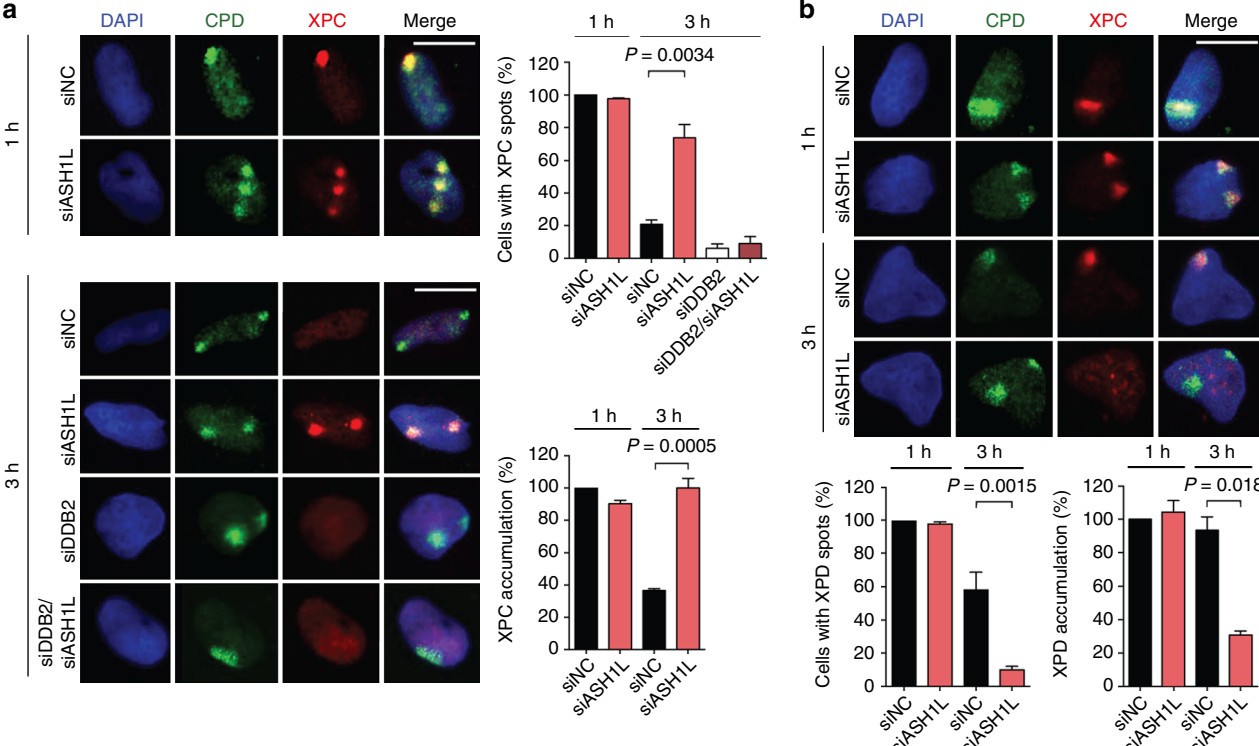

**Fig. 5** Dysregulated GG-NER assembly upon ASH1L depletion. **a** Retention of XPC protein in CPD spots of ASH1L-depleted cells. Transfections with siRNA occurred 48 h before UV radiation (100 J m$^{-2}$) through 5-μm filter pores. Immunofluorescence was assessed at the indicated times after treatment. Quantifications show the percentage of CPD-positive cells containing visible protein spots, and ratios of fluorescence intensity of each protein spot divided by the nuclear background ($n = 3$ with at least 100 cells in each experiment). The intensity ratio seen with siNC controls is set to 100%; error bars show s.e.m. (one-sample t-test). Scale bar = 15 μm. **b** Impaired relocation of XPD to the CPD spots of ASH1L-depleted cells. Quantifications were based on at least 100 cells per experiment ($n = 3$, two-tailed t-test).

particles devoid of H3K4me3 (Fig. 7a). The level of XPC and DDB2 bound to core particles was normalized to the total content of histone H3 in each precipitated fraction, revealing that XPC protein displays a marked preference for core particles containing H3K4me3, whereas DDB2 exhibits an opposite bias, i.e., for core particles lacking H3K4me3 (Fig. 7b).

It remained possible that the interaction of XPC protein with core particles is mediated entirely by its association with DNA. To rule out this possibility, we digested the MNase-liberated core particles with benzonase, which is commonly used for the removal of nucleic acids from biological matrices. This degradation of DNA wrapped around histone octamers in core particles did not reduce the amount of associated XPC protein. On the contrary, the benzonase digestion increased the interaction of XPC protein with DNA-free octamers precipitated with anti-H3K4me3 antibodies, indicating an intrinsic ability of XPC protein to interact with histones (Fig. 7c, d). This finding was confirmed by demonstrating that immunoprecipitation of MNase- and benzonase-digested chromatin with anti-XPC antibodies leads to the co-isolation of histone octamers containing H3K4me3 (Fig. 7e).

To investigate the hypothesis of a direct XPC-histone interaction, recombinant XPC protein expressed in insect cell (sf9) lysates was incubated for 1 h with purified recombinant H3K4me3. The samples were subjected to immunoprecipitation using anti-H3K4me3 antibodies and the isolates analyzed for their XPC content (Fig. 7f). This proven co-immunoprecipitation of XPC with H3K4me3, using recombinant factors, supports the conclusion that the XPC subunit has an intrinsic affinity for core histones.

**Identification of an H3K4me3-interacting domain**. The enhanced affinity of XPC protein for DNA-free histone octamers, compared to DNA-bound octamers (Fig. 7c), suggested the existence of a histone-binding motif that is partially masked by interactions of neighboring domains with the DNA substrate. Between the two well-characterized β-hairpin domains BHD2 and BHD3, responsible for the DNA-binding activity of human XPC protein, we identified a short β-turn motif (residues 741–757) that does not make contacts with DNA[59, 60], but displays negatively charged amino acids that may interact with the positively charged histones (Fig. 8a). We predicted that it should be possible to reduce the affinity of XPC protein for histones by replacing negatively charged amino acids in this motif with positively charged analogs. The consequence of D748K and E755K charge inversions was tested by expressing green-fluorescent protein (GFP) fusions containing single and double mutations in XP-C fibroblasts, and using 0.3-M NaCl extracts to test their interaction with recombinant H3K4me3. Immunoprecipitations with anti-H3K4me3 antibodies, followed by analysis using anti-GFP antibodies, demonstrated that the D748K mutation (alone or together with the E755K change) reduces the association of XPC protein with H3K4me3 (Fig. 8b). Remarkably, the residual binding of these charge inversion mutants with H3K4me3 is weaker than that observed for the W690S mutant with an overall disrupted protein folding and stability[61]. These GFP constructs were used to complement the GG-NER defect of UV-irradiated XP-C fibroblasts. A comparison of CPD excision determined by immunoassay showed that wild-type XPC protein, but not the W690S mutant, is able to complement the GG-NER defect of XP-C cells. This assay allowed us to show that the reduced histone-binding

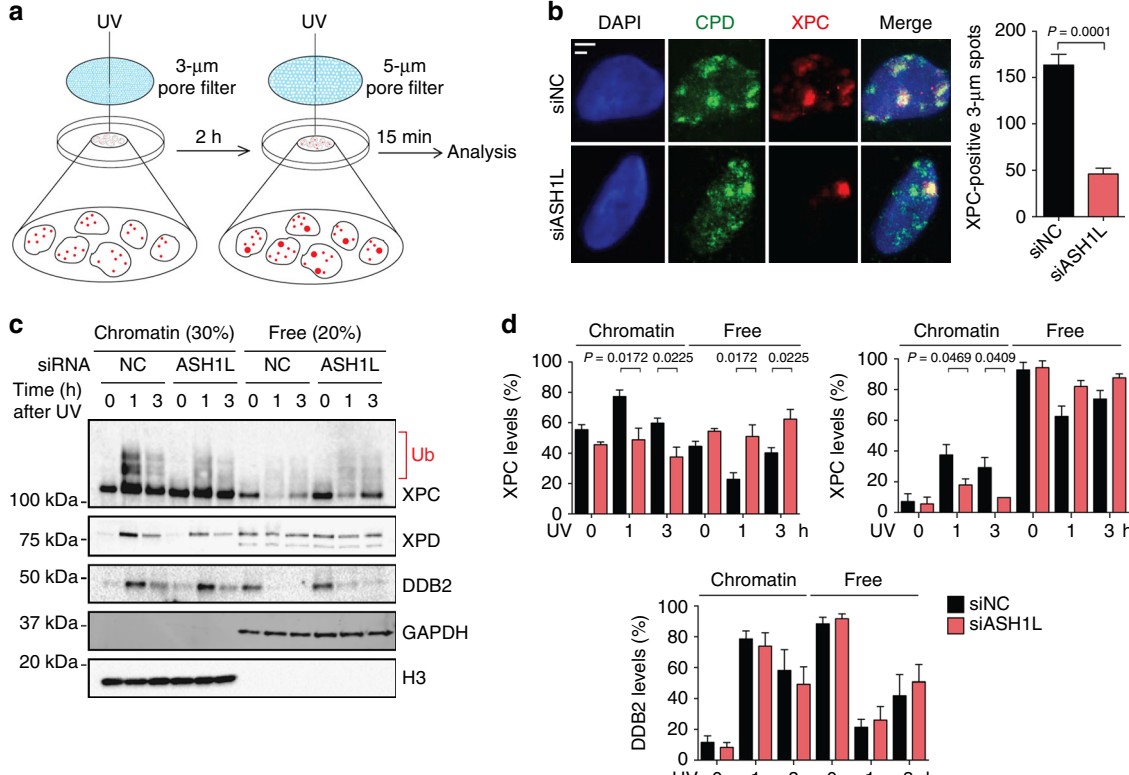

**Fig. 6** Reduced binding of XPC to CPD sites upon ASH1L depletion. **a** Scheme illustrating the experimental strategy to monitor the dynamics of XPC protein in the chromatin of living cells. At the time of analysis, the cells display small UV spots (generated with 3-μm pore filters) containing only CPDs and large UV spots (generated with 5-μm pore filters) containing a mixture of both 6–4PPs and CPDs. **b** ASH1L depletion compromises the ability of XPC protein to associate with the CPDs in small spots. Transfections with siRNA occurred 48 h before UV radiation of U2OS cells through 3-μm filter pores. Immunofluorescence was assessed 15 min after the second UV treatment through 5-μm filter pores. Scale bars = 3 μm and 5 μm. Quantifications show the amount of cells containing small XPC spots (n = 4 with 200 cells in each experiment); error bars show s.e.m. (two-tailed t-test). **c** Binding of XPC, XPD and DDB2 to chromatin upon UV radiation. HeLa cells were transfected with siASH1L or siNC, exposed to UV light (10 J m$^{-2}$) 48 h later, and analyzed by immunoblotting at different times. Chromatin was salt-extracted to release free proteins. GAPDH and H3 were the loading controls for free proteins and chromatin, respectively. **d** Level of XPC, XPD and DDB2 in chromatin, relative to the respective free proteins, normalized to H3 and GAPDH (n = 4 independent experiments, two-tailed t-test). These quantifications include both unmodified and ubiquitinated XPC protein and, by showing relative levels, take into account the overall higher XPC content of ASH1L-depleted cells

capability of the D748K mutant correlates with impaired CPD repair (Fig. 8c).

To confirm that the β-turn motif of human XPC participates in histone binding, we generated polypeptides XPC$_{607-741}$, containing only BHD1 and BHD2, and XPC$_{607-766}$, containing the β-turn motif in addition to BHD1 and BHD2 (Fig. 8d). In co-immunoprecipitations, the longer fragment XPC$_{607-766}$ retained the ability to associate with H3K4me3, whereas this histone-binding activity was essentially lost with the shorter fragment XPC$_{607-741}$ (Fig. 8e). We conclude that the β-turn motif of human XPC protein contributes to its association with the core histones of nucleosomes and that this interaction determines the efficiency of CPD excision.

## Discussion

There are several potential mechanisms by which histone methyltransferases may participate in the cellular UV response. For example, the methylation of histone H3 at position K79 by DOT1L (for *Disruptor Of Telomeric silencing 1-Like*) has been shown to support transcription upon genotoxic stress without influencing DNA repair activity[62]. By contrast, we describe the ASH1L histone methyltransferase as an auxiliary factor that, like DDB2 (the product of the *XPE* gene), is indispensable for excision

of CPD lesions. Although DDB2 is not a core subunit of the GG-NER system, its absence or functional defect in XP-E cells causes reduced CPD repair[20, 63]. The generally accepted model is that DDB2 protein recognizes CPDs and delivers these lesions to XPC, which is the initiator of GG-NER activity[53]. However, the mechanism of this handover remained elusive because reconstitution assays showed that DDB2 is not needed for CPD excision from naked DNA[64] and because it is not possible, in biochemical experiments using histone-free DNA substrates, to detect and characterize stable intermediates where DDB2 and XPC bind to the same damage simultaneously[16]. It is, therefore, assumed that DDB2 deploys its role within the chromatin context where it undergoes transient interactions with XPC regulated by CUL4A-mediated ubiquitination[23, 65, 66]. Ubiquitination of XPC protein increases its DNA-binding affinity[27] and the concomitant ubiquitination of histones is thought to facilitate access to the DNA substrate[67, 68].

We present the histone methyltransferase ASH1L as a so far unknown missing link between DDB2 and XPC during initiation of the GG-NER process (Fig. 9). DDB2 bound to UV lesions in nucleosome cores recruits the histone methyltransferase ASH1L. The CUL4A ubiquitin ligase, to which DDB2 binds through its DDB1 adapter, is not required for this ASH1L recruitment. In response to UV damage, the DDB2-ASH1L interaction (Fig. 1g, h) leads to methylation of lysine 4 in histone H3. In turn, the

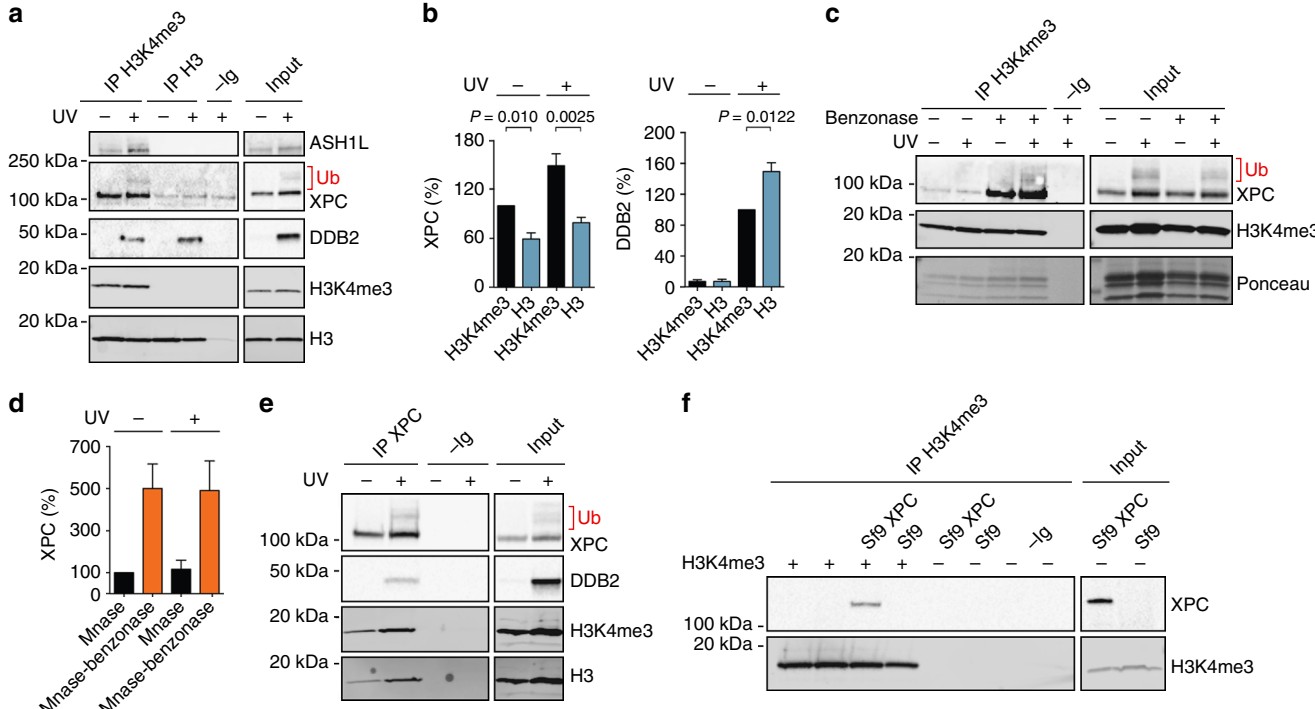

**Fig. 7** Association of XPC with nucleosomes containing H3K4me3. **a** Interaction of XPC protein with core particles generated by MNase digestion of chromatin. These core particles were immunoprecipitated with anti-H3 or anti-H3K4me3 antibodies and analyzed by immunoblotting; –Ig, immunoglobulins omitted (core particles from UV-irradiated cells subjected to the same procedure except that antibodies were left out). **b** Quantification of XPC associating with core particles precipitated with anti-H3 or anti-H3K4me3 antibodies. The UV dose was 10 J m$^{-2}$; error bars show s.e.m, ($n = 3$, one-sample t-test with a hypothetical value of 100). **c** Interaction of XPC protein with histone octamers generated by MNase and benzonase digestion of chromatin. These octamers were immunoprecipitated with anti-H3K4me3 antibodies and analyzed by immunoblotting; –Ig, immunoglobulins omitted. Histones were stained with the Ponceau dye. **d** Quantification of XPC associating with histone octamers precipitated with anti-H3K4me3 antibodies ($n = 3$, one-sample t-test). XPC levels in the unirradiated controls are set to 100%. **e** Histone octamers obtained by MNase and benzonase digestion of chromatin were immunoprecipitated with anti-XPC antibodies; –Ig, immunoglobulins omitted. **f** Interaction between recombinant XPC and H3K4me3. Lysates from sf9 cells expressing XPC protein were supplemented with H3K4me3 (2 μg) as indicated and immunoprecipitated using anti-H3K4me3 antibodies

methylated histone H3K4me3 allows for the stable docking of the XPC complex to the histone octamer of UV-damaged nucleosomes. In the absence of ASH1L, XPC protein remains only weakly bound to the nucleosome through its association with DDB2. By favoring the formation of a stable intermediate of XPC with the methylated histone octamer, ASH1L facilitates the recruitment of TFIIH and downstream factors of the GG-NER pathway. Without ruling out the possibility that multiple XPC domains may interact with histone octamers, we found that the β-turn motif situated between the known DNA-binding domains BHD2 and BHD3 contributes to the association of XPC protein with H3K4me3.

Against the generally contended notion that the XPC subunit is unable to access histone-bound substrates, we show that the histone octamer of DNA-damaged nucleosomes provides, instead, an interaction platform for the critical substrate handover from DDB2 to the XPC partner, thereby initiating the GG-NER reaction at CPD sites in chromatin. That the association of XPC protein with H3K4me3 is functionally important can be inferred from the finding of Fig. 8c, whereby an XPC mutant that interferes with XPC-H3K4me3 interactions reduces GG-NER activity in living cells. As observed for DDB2 defects, down regulation of ASH1L confers an exquisitely slow excision of CPDs and, in addition, causes UV hypersensitivity. That ASH1L mutations have not been identified in the context of the XP syndrome may be due to the observation that inactivation of this essential histone methyltransferase is not compatible with developmental processes.

## Methods

**Cell lines**. HeLa and U2OS cells were obtained from American Type Culture Collection. SV-40-transformed XP-C fibroblasts (carrying a homozygous C-to-T transition at codon 718) were from Coriell Institute of Medical Research. All cell lines were grown using Dulbecco's modifies Eagle medium supplemented with 10% (v/v) fetal calf serum (FCS, Gibco), 100 U ml$^{-1}$, penicillin and 100 μg ml$^{-1}$ streptomycin. Although these cells do not belong to the commonly misidentified lines, they were authenticated by short tandem repeat (STR) profiling (HeLa and U2OS) or isoenzyme electrophoresis (fibroblasts) and tested negative for mycoplasma contamination.

**Protein depletion**. The siRNA reagents are listed in Supplementary Table 1. Transfections were performed with Lipofectamine RNAiMAX (Thermo Scientific) according the manufacturer's protocol. SiRNA concentrations were 16 nM for the silencing of XPC, DDB2, CUL4A, XPA, CSB and SETD2, 20 nM for the silencing of ASH1L. Experiments were carried out 48 h after siRNA transfections.

**UV irradiation**. Irradiation with UV-C light was performed with a germicidal lamp (wavelength 254 nm) after washing the cells with phosphate-buffered saline (PBS). Local damages were generated by irradiation with 100 J m$^{-2}$ through a 3-μm or 5-μm pore polycarbonate filter (Millipore). After UV treatment, cells were incubated with complete culture medium before processing for further analyses.

**Plasmid transfection**. Transfections with XPC-GFP expression vectors were carried out using FuGENE HD (Promega) reagent according to manufacturer's protocol. Cell were transfected at 80% confluency as described[69] and lysed 24 h later.

**Gene expression**. Total RNA was extracted from cells with the RNase isolation kit (Qiagen), according to the manufacturer's protocol. DNA was digested using DNase I, and RNA concentration was determined using the NanoDrop instrument. Briefly, 500 ng of RNA from each sample where subjected to reverse transcription according to manufacturer's protocol (iScript cDNA synthesis Kit, Bio-Rad).

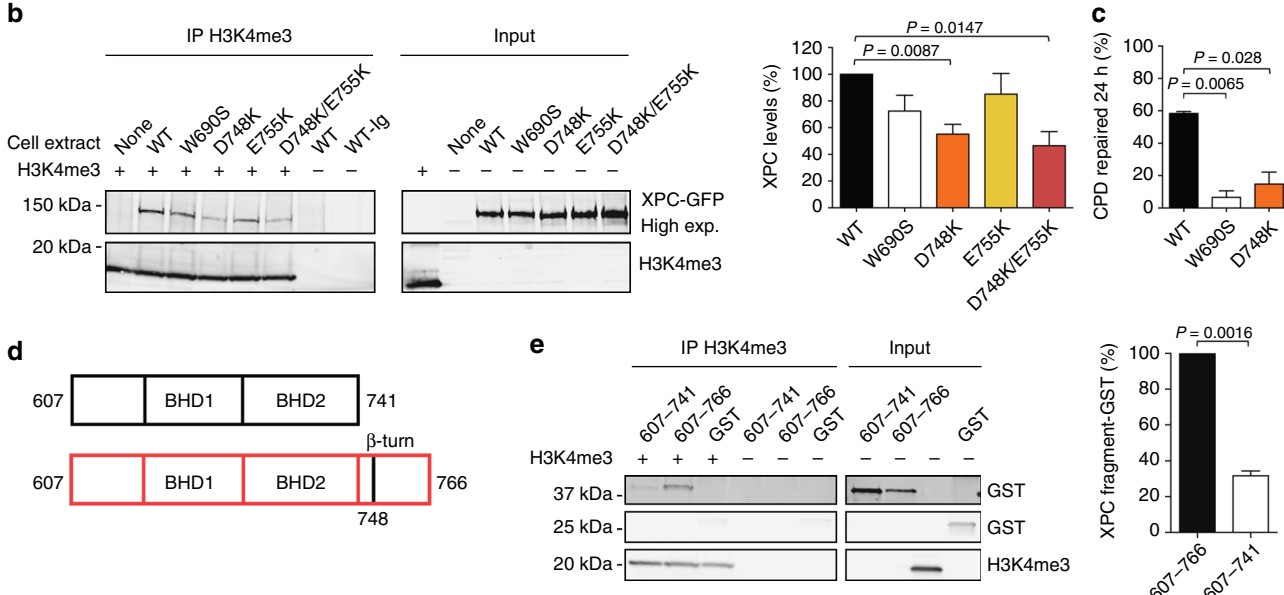

Fig. 8 Histone-interacting domain in human XPC protein. **a** Sequence of the β-turn in human XPC protein showing the introduced charge inversions. **b** Association of wild-type (WT) or mutant XPC with H3K4me3. The indicated GFP fusions were expressed in XP-C fibroblasts and 0.3-M NaCl extracts were incubated with recombinant H3K4me3 (2 μg), followed by immunoprecipitation using anti-H3K4me3 antibodies. Quantifications of co-immunoprecipitated XPC, shown as the percentage of wild-type controls, were carried out relative to the respective input. Error bars indicate s.e.m ($n = 3$, one-sample t-test). **c** Excision of CPDs in XP-C fibroblasts transfected 24 h before UV radiation (10 J m$^{-2}$) with vectors coding for wild-type (WT) XPC or the indicated mutants ($n = 3$, each experiment with 4 replicates, two-tailed t-test). Immunoassays for CPD quantifications were carried out after a 24-h repair incubation. **d** Domains in polypeptide fragments XPC$_{607-741}$ and XPC$_{607-766}$. **e** The indicated XPC polypeptides fused to glutathione-S-transferase (GST; 4 μg) and GST alone (2.5 μg) were incubated with H3K4me3 (2 μg), immunoprecipitated with anti-H3K4me3 antibodies and visualized using anti-GST antibodies. Quantification of co-isolated polypeptides relative to the respective input ($n = 3$, one-sample t-test). Values obtained with the larger fragments are set to 100%

Quantitative RT-PCR was performed using the KAPA SyBR FAST Universal qPCR Kit (KAPA BIOSYSTEMS) according to the manufacturer's protocol. Reactions were carried out in duplicates using the CFX384 Real-time System C1000 Touch Thermal Cycler. Relative gene expression levels were presented as $2^{-\Delta\Delta CT}$, normalized against GAPDH and Beta-2 microglobulin (B2M), and presented relative to controls treated with non-coding siRNA. The Quantitative RT-PCR primers were obtained from Microsynth (Supplementary Table 2).

**Immunofluorescence.** Glass coverslips of 12 mm (Thermo Scientific) were used to grow U2OS cells to 80% confluency and irradiated through 3-μm or 5-μM filter pores to induce local UV damage. After the indicated times, the culture medium was removed and a pre-extraction buffer [25mM HEPES, pH7.5, 50 mM NaCl, 1 mM EDTA, 3 mM MgCl$_2$, 300 mM sucrose and 0.5% (v/v) Triton X-100] added for 2.5 min at 4 °C. Thereafter, cells were fixed with 4% (w/v) paraformaldehyde (pH 8.0) for 15 min and permeabilized for 20 min with PBS containing 0.05% (v/v) Tween 20. PBS with 20% FCS was used for a blocking step of 30 min at 37 °C. Primary antibodies (Supplementary Table 3) were diluted in PBS with 5% FCS and applied for 1 h at 37 °C. Secondary antibodies (Supplementary Table 3), diluted in PBS with 5% FCS, were added for 30 min at 37 °C after washing with PBS-Tween 20. DNA was stained with DAPI (0.2 μg ml$^{-1}$). Immunofluorescence images were taken with a fluorescence inverted microscope (Leica, DMI6000 B, 63x oil Plan-Apochromat, 1.4 numerical aperture oil immersion lens) and analyzed with ImageJ software. Cell numbers displaying the indicated spots were determined by counting all CPD-positive cells with the respective protein staining. The quantified protein levels at UV lesion sites were expressed as the ratio of fluorescence intensity at the damage spot and the respective intensity in the remaining nuclear area after subtraction of the background intensity outside the nucleus.

**Immunoblotting.** Cells were treated as indicated, washed with PBS and lysed as indicated. Protein concentrations were measured by Pierce BCA Protein Assay Kit (Thermo Scientific) at 562 nm. Loading buffer was added to 60 mM Tris-HCl, pH 6.8, 10% (v/v) glygerol, 2% (w/v) sodium dodecyl sulfate (SDS), 1.25% (v/v) β-mercaptoethanol, 0.01% (w/v) bromophenol blue (final concentrations) and the samples heated for 10 min to 95 °C. In each case, 2–50 μg of sample proteins were separated on 4–20% Criterion TGX stain-free precast gels (Bio-Rad) for 30 min at 250 V and transferred to nitrocellulose membranes using the Turbo transfer device (Bio-Rad, 7 min at 5 A). The signals resulting from antibody incubations were documented and quantified with the Odyssey CLx Imaging System (LI-COR) or the Chemidoc MP Imaging System (Bio-Rad). Uncropped versions of all blots are shown in Supplementary Figs. 7, 8 and 9.

**Chromatin digestion.** Confluent cell layers in 10-cm dishes were UV-irradiated and lysed by incubation (30 min on ice using a turning wheel) with NP-40 buffer [25 mM Tris-HCl, pH 8.0, 0.3 mM NaCl, 1 mM EDTA, 10% (v/v) glycerol, 1% (v/v) NP-40, 0.25 mM phenylmethylsulfonyl fluoride and EDTA-free protease inhibitor cocktail (Roche)][58]. Next, free proteins not bound to chromatin were separated by centrifugation at 15,000 g for 10 min. The remaining chromatin pellet was suspended in CS buffer [20 mM Tris-HCl, pH 7.5, 100 mM KCl, 2 mM MgCl$_2$, 1 mM CaCl$_2$, 0.3 M sucrose and 0.1% (v/v) Triton X-100] and supplemented with reaction buffer containing, as final concentrations, 50 mM Tris-HCl, pH 7.9, 5 mM CaCl$_2$, bovine serum albumin (100 μg ml$^{-1}$) and 0.4 U ml$^{-1}$ MNase (New England Biolabs). After a 20-min incubation at 37 °C, the reaction was stopped with EDTA (5 mM) and the solubilized chromatin supernatant was collected after centrifugation 15,000 g for 10 min.

**Quantification of UV lesions.** Enzyme-linked immunosorbent assays were carried according to the manufacturer's instructions to quantify UV lesions. Genomic DNA was extracted using the DNeasy blood and tissue Kit (Qiagen) and heat-denatured at 99 °C for 10 min followed by a 15-min incubation on ice. The 96-well microtiter plates (Greiner) were coated with protamine sulfate (Sigma), dried overnight at 37 °C and loaded with 50 μl of DNA solution per well (4 μg ml$^{-1}$ for 6–4PP detection, 200 ng ml$^{-1}$ for CPD detection). Once coated with DNA, the

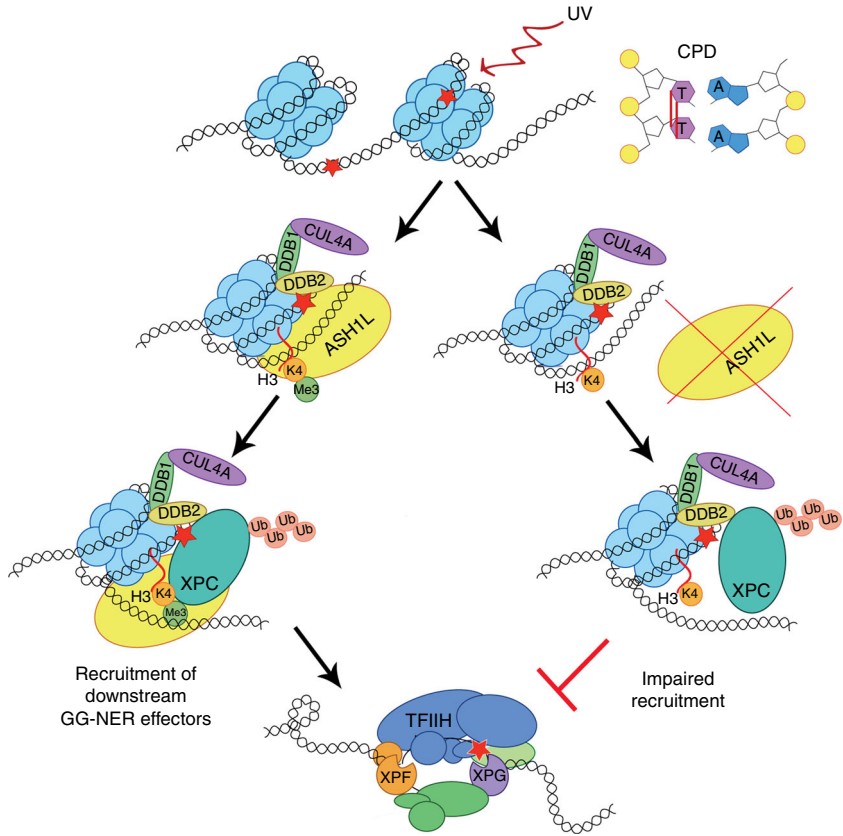

**Fig. 9** Model of CPD recognition in nucleosome cores. UV irradiation induces CPDs that are distributed evenly in chromatin and, hence, are formed also within nucleosome cores where the DNA is wrapped around histone octamers. Recognition of CPDs in nucleosome cores by DDB2 leads to recruitment of the histone methyltransferase ASH1L and, as a consequence, to methylation of the amino-terminal tail of histone H3 at position lysine 4. The presence of methylated H3 facilitates the stable docking of the XPC subunit to damaged sites in the nucleosome core, such that XPC protein is endowed with the ability to induce downstream GG-NER reactions, including DNA unwinding by TFIIH and dual DNA incision by the XPG and XPF-ERCC1 endonucleases. In the absence of ASH1L and H3K4 methylation, the loose interaction between DDB2 and XPC is not sufficient to initiate the GG-NER reaction at nucleosomal CPDs

plates were washed five times with PBST [0.05% (v/v) Tween 20 in PBS] and blocked with 2% (v/v) FCS in PBS at 37 °C for 60 min. Next, the plates were incubated with antibodies against 6–4PPs (64M-2; dilution of 1:2000) and CPDs (TDM-2; dilution of 1:5000) for 30 min at 37 °C. These primary antibodies were detected by biotin-labeled F(ab')2 fragments obtained from anti-mouse IgG (dilution 1:2000; see Supplementary Table 3) added for 30 min at 37 °C. After washing the plates, 100 μl of a peroxidase streptavidin conjugate (1:10,000) were added to each well. The color reaction was started with 0.5 mg ml$^{-1}$ o-phenylenediamine, 0.007% (v/v) $H_2O_2$ and citrate-phosphate buffer (50 mM $Na_2HPO_4$, 24 mM citric acid, pH 5.0). After stopping the reaction with 50 μl of 2 M $H_2SO_4$, absorbance was detected at 492 nm in a PLUS384 microplate spectrophotometer.

**Analysis of MNase-digested DNA**. Cells were lysed on ice 30 min with NP-40 buffer. After a 10-min centrifugation at 15,000 g, the remaining pellet was resuspended in CS buffer, supplemented with reaction buffer and MNase-digested as outlined above. Control samples were incubated without MNase enzyme. The DNA was extracted by adding TE buffer (10 mM Tris-HCl, pH 8.0, 1 mM EDTA) and neutral phenol. After 15 min of shacking, samples were centrifuged (5 min at 6000 g), the phenol was discarded and the aqueous solution washed twice with chloroform. The DNA was precipitated in ethanol supplemented with 100 mM sodium acetate, dried and resuspended in TE buffer. DNA concentration was measured using the NanoDrop device; 0.5 μg of DNA were separated on a 1.75% (w/v) agarose gel.

**Unscheduled DNA synthesis**. Repair patch synthesis was monitored by immunofluorescence[54]. U2OS cells were cultured on 12-mm coverslips and locally (100 J m$^{-2}$) UV-irradiated through 5-μm filter pores. After irradiation, cells were incubated for 2 h in culture medium and then another 1 h in culture medium supplemented with 10 μM 5-ethynyl-2'-deoxyuridine (EdU; Thermo Scientific). Thereafter, the cells were washed with PBS, pre-extracted for 2.5 min and fixed with 4% (w/v) paraformaldehyde (pH 8.0) at room temperature for 15 min followed by permeabilization for 20 min with PBS containing 0.05% (v/v) Tween 20. Before blocking in PBS with 20% FCS for 30 min at 37 °C, the DNA was denatured

for 8 min in 0.07 M NaOH. Antibodies against CPDs were added for 1 h at 37 °C. Cells were then washed for 20 min with PBS containing 0.05% (v/v) Tween 20. DNA was stained with DAPI (0.2 μg ml$^{-1}$) and secondary antibody. Incorporated EdU was coupled to Alexa Fluor 488 using Click-iT EdU Imaging Kit (Thermo Scientific) following the manufacturer's instructions. Images of immunostained cells were taken with a fluorescence inverted microscope (Leica, 63x oil Plan-Apochromat, 1.4 numerical aperture oil immersion lens) and analyzed with ImageJ software. EdU incorporation was analyzed in 100 cells by determining fluorescence intensity in the UV-damaged areas (CPD spots) divided by the background nuclear intensity after subtraction of the background intensity outside the nucleus. S-phase cells displaying high EdU fluorescence in their entire nucleus were excluded from the analysis.

**Recovery of RNA synthesis**. RNA synthesis was monitored by immunofluorescence[54]. U2OS cells were cultured on 12-mm coverslips and globally (16 J m$^{-2}$) UV- irradiated. After irradiation, cells were incubated for 4 h in culture medium and then 2 h in culture medium supplemented with 100 μM 5-ethynyl uridine (EU; Thermo Scientific). Thereafter, the cells were washed with PBS, pre-extracted for 2.5 min and fixed with 4% (w/v) paraformaldehyde (pH 8.0) at room temperature for 15 min followed by permeabilization for 20 min with PBS containing 0.05% (v/v) Tween 20. A blocking step was carried out in PBS with 20% FCS for 30 min at 37 °C. Cells were then washed for 20 min with PBS containing 0.05% (v/v) Tween 20. DNA was stained with DAPI (0.2 μg ml$^{-1}$). Incorporated EU was coupled to Alexa Fluor 488 using the Click-iT Alexa Fluor 488 Imaging Kit (Thermo Scientific) following the manufacturer's instructions. Images of immunostained cells were taken with a fluorescence inverted microscope (Leica, 63x oil Plan-Apochromat, 1.4 numerical aperture oil immersion lens) and analyzed with ImageJ software. EU incorporation was monitored in 100 cells by determining the nuclear fluorescence intensity after subtraction of the background intensity outside the nucleus.

**UV sensitivity**. To measure viability, 300,000 HeLa cells were siRNA-transfected as indicated above in 6-well plates. Following 48 h after transfection, 5000 cell in four replicates for each condition were trypsinized and seeded in 96-well plates.

After 24 h, cells were UV-irradiated with 0, 2.5, 5 or 10 J m$^{-2}$. After another 24 and 48 h, the culture medium was replace with 100 µl medium containing 10% (v/v) AlamarBlue solution (Sigma). Sample fluorescence was measured after 3-hour incubations using the fluorospectrometer LS-55 from Perkin Elmer set to EX = 560 and EM = 590.

**Immunoprecipitation of endogenous proteins**. After two washing step on ice with PBS, 10 µl of slurry Protein G sepharose (GE Healthcare) for each sample were incubated with the indicated antibodies (Supplementary Table 3) on a turning wheel for 45 min at 4 °C. After centrifugation (1 min, 100 g), protein G-sepharose was suspended in buffer A [0.5 M Tris-HCl, pH 8.0, 8% (v/v) glycerol, 300 mM NaCl, 2% (v/v) Triton X-100, 2 mM EDTA, 0.25 mM phe-nylmethylsulfonyl fluoride and ETDA-free protease inhibitor cocktail (Roche)]. Next, 100 µl of protein G-sepharose were added to 50 µg cell extracts and incubated 3 h at 4 °C on a turning wheel. Beads were washed twice by centrifugation (2 min, 100 g) in HNTG buffer [20 mM HEPES, pH 7.5, 150 mM NaCl, 0.1% (v/v) Triton X-100 and 10% (v/v) glycerol]. Protein elution was carried out by boiling the sample to 95 °C in loading buffer [60 mM Tris-HCl, pH 6.8, 10% (v/v) glycerol, 2% (w/v) sodium dodecyl sulfate (SDS), 1.25% (v/v) β-mercaptoethanol, 0.01% (w/v) bromophenol blue]. Samples were analyzed by polyacrylamide gel electrophoresis and immunoblotting as described above.

**Chromatin immunoprecipitation**. Proteins were crosslinked to DNA by addition of 1% (v/v) formaldehyde for 10 min on ice; 0.125 M of glycine was added to stop the reaction. After two washing steps on ice with PBS, cells were lysed and chromatin was MNase-digested as outlined above. Ten µl of Protein G sepharose were incubated for 45 min at 4 °C on a turning wheel with 5 µl anti-DDB2 antibodies. After centrifugation (1 min, 100 g), protein G-sepharose was suspended in buffer A. Next, 100 µl of protein G-sepharose were added to 150 µg solubilized chromatin and incubated 3 h at 4 °C on a turning wheel. Beads were then washed in HNTG buffer. Protein elution was carried out by boiling the sample to 95 °C in loading buffer. Samples were analyzed by polyacrylamide gel electrophoresis and immu-noblotting as described above.

**Immunoprecipitation of recombinant histones**. Protein G sepharose (GE Healthcare) was prepared as above and incubated with anti-H3K4me3 antibodies (Supplementary Table 3) on a turning wheel for 45 min at 4 °C. Recombinant histone H3K4me3 (2 µg) (Active Motif), were incubated for 1 h at 4 °C on a turning wheel directly with 50 µg of cell extracts. After centrifugation (1 min, 100g), protein G-sepharose was suspended in buffer A; 100 µl of protein G-sepharose were added to the cell extract containing recombinant histones and incubated 3 h at 4 °C on a turning wheel. Beads were then washed twice by centrifugation (2 min, 100g) in HNTG buffer. Protein elution, polyacrylamide gel electrophoresis and immuno-blotting were carried out as described above.

**Co-immunoprecipitation of XPC-GST**. Slurry Protein G sepharose (GE Health-care) was prepared as described above and incubated with the anti-H3K4me3 antibody (Supplementary Table 3) on a turning wheel for 45 min at 4 °C. Recombinant histone H3K4me3 (2 µg) (Active Motif), were incubated for 1 h at 4 °C on a turning wheel together with XPC-fragments (4 µg) or purified GST (2.6 µg) (Sigma) in YY buffer [100 mM HEPES, pH 7.5, 20% (v/v) glycerol, 2% (v/v) tryton, 2 µM ETDA]. After centrifugation (1 min, 100 g), protein G-sepharose was suspended in buffer A; 100 µl of protein G-sepharose were added to 50 µl YY buffer containing recombinant histones and XPC-GST or purified GST. After 3 h incubations at 4 °C on a turning wheel, beads were washed twice by centrifugation (2 min, 100 g) in HNTG buffer. Protein elution, polyacrylamide gel electrophoresis and immunoblotting were carried out as described above.

**Statistical analyses**. Data are shown as mean values with s.e.m. Differences were calculated in GraphPad Prism 6 using the statistical tests indicated in the figure legends and the resulting P values are given directly in the figures. In all experi-ments, between-group variances were similar and data were symmetrically dis-tributed. In immunofluorescence studies, at least 100 random cells per experiment were scored in each group.

**Data availability**. All relevant data are available from the authors.

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

## Acknowledgements

This work was supported by the Velux Stiftung (Project 753), the Swiss Cancer League (2832–02–2011) and the Swiss National Science Foundation (Grant 31003A_170111/1 to H.N. and 31003A_166370/1 to L.P.). We also acknowledge support by the Center of Clinical Studies.

## Author contributions

C.B.P., M.G. and H.N. devised and planned the experiments, C.B.P., P.R. and Z.G. carried out the experiments and analyzed the data, C.B.P., L.P. and H.N. wrote the manuscript.

## Additional information

**Competing interests:** The authors declare no competing financial interests.

