## [Peer Review File · Nature Communications]

Reviewers' comments:

Reviewer #1 (Remarks to the Author):

This is a very interesting manuscript that describes a potentially novel mechanism by which DDB2 facilitates XPC (the initiator of GG-NER) recruitment and binding to DNA damage in the context of chromatin. The authors identify Histone methyltransferase ASH1L as novel UV-response protein by means of a dedicated Histone methyltransferase siRNA screen. Based on their experiments, they conclude that ASH1L is recruited to DNA damage through DDB2 to methylate H3K4 which promotes stable association of XPC to UV-damaged nucleosomes. They also showed that XPC interacts preferentially with methylated H3K4 via negatively-charged residues in a beta-turn motif adjacent to known DNA-interacting domains. ASH1L is required for efficient removal of CPDs, but not 64PP, by GG-NER.

In spite of the fact that this manuscript presents exciting data and a novel concept of successive protein handover at the initial steps of GG-NER, there are several issues that first need to be resolved before this work is sufficiently strong to be acceptable for publication in Nature communications.

Major issues.

1. In the Histone methyltransferase siRNA screen (suppl Figure 1), the authors observed the strongest effect upon knock down of ASH1L. However, also depletion of other Histone methyltransferases reduced the cellular viability after UV. The authors should comment on this. It is very well possible that not only ASH1L is involved in the UV-response but that other methyltransferases act (partial) redundant.

Moreover in the introduction part of the first results paragraph the authors ignored the previous findings by Oksenyich et al (group of Coin, PLoS Gen, 2013) that DOT1L depletion confers UV-hypersensitivity. Discussion of this publication should be included.

2. Fig 1E, F suggests that chromatin recruitment of ASH1L depends on DDB2 but not CUL4A. However, DDB2 recruitment to chromatin is reduced upon CUL4A depletion, suggesting that siCUL4A should also affect ASH1L recruitment. The authors should address this apparent inconsistency.

3. lines 182-184 and Figure 3b. The applied assay for transcription recovery after UV seems rather strange and the presented data could not have been derived with the here described procedure. To measure recovery of RNA synthesis, the authors added EU for 1 hour immediately after UV irradiation. After UV, transcription levels decrease for multiple hours, also in TC-NER-proficient cells. Efficient TC-NER-dependent transcription recovery is only measurable after a longer time, usually more than 16 hours (with the here applied UV-dose of 10 J.m⁻²). To measure this recovery, EU levels before and following an extensive recovery period (> 16 hours) after UV should be compared.

Importantly, shortly after UV transcription also declines in TC-NER-proficient cells, usually to the same extent as in TC-NER-deficient cells. However, transcription will only resume in TC-NER proficient cells after extended incubation times (>16 hour following 10 J/m²). The authors should properly carry out this experiment to be able to make any definite conclusions on the involvement of ASH1L in TC-NER. Also, it is unclear which cells were used in Figure 3a and b?

4. Page 6-7; Figure 4D. The quantification in this Figure 4D is unclear and it is not convincingly shown that ASH1L is required for XPC recruitment. The relative amounts of XPC in the chromatin fraction compared to the free fraction were quantified from the blot shown in Figure 4C and these quantifications suggest that after siASH1L, XPC does not increase in the chromatin fraction. However, in the western blot, also for siASH1L, there is a clear increase of XPC in the chromatin fraction and a concomitant strong decrease in the free fraction after UV. Also, the quantification does not take into account that XPC levels are higher for siASH1L. This should be clarified and/or

adjusted.

The authors also argue that if XPC levels are higher after siASH1L, less extra recruitment of XPC may be needed. This could indeed be the case and the fact that less XPD is recruited could also mean that ASHL1 promotes TFIIH recruitment instead of XPC. This possibility, also suggested by the IF experiments in Figure 5, should be considered and addressed by the authors.

5. One of the main conclusions of this manuscript, that ASHL1 activity promotes XPC recruitment, is contradicted by the IF experiments shown in Figure 5. Rather, these experiments suggest that ASHL1 activity promotes release of XPC and/or recruitment of TFIIH. The authors recognize the discrepancy with the results shown in Figure 4 but insufficiently try to explain this and ignore it in the remainder of the manuscript and in the discussion. The authors suggest that the difference is due to more stringent extraction procedures in the chromatin fractionation. However, this still does not satisfactorily explain why XPC recruitment in the IFs is prolonged after siASH1L. The authors should try to better reconcile these two types of experiments or change their conclusions. If the authors think that ASHL1 only promotes the stable association of XPC with damage but not its DDB2-dependent recruitment, then they should prove this idea in a more straightforward manner. If ASHL1 functions in a more stable association (reduced extractability) of XPC to damaged chromatin, this could be directly measured by FRAP on UV-damaged cells or UV-damaged nuclear areas, as was previously nicely shown by the same group (Camenish et al, EMBO J, 2009) that this procedure reveals binding properties of XPC to damaged DNA (chromatin).

6. The authors should test the functional significance of their findings in Figures 6 and 7 that XPC interacts with H3K4me3 via a beta-turn motif, by testing whether the D748K mutant XPC displays altered recruitment kinetics and/or whether repair efficiency is affected by this mutation.

Minor issues

7. Figure 2A. This important figure, which is used to show that UV-induced H3K4 methylation depends on ASHL1, would better fit with the results shown in Figure 1. The authors should also include the 1 and 3 hour time point for the NC, so that the increase in H3K4me3 in the control can be compared. Also, to make a really strong case, the authors could confirm that this all depends on DDB2 by including siDDB2.

8. Figure 2G. CPD removal as measured by immunofluorescence in U2OS cells seems much faster (remarkable reduction of 80% of the signal as compared to immediately after UV) than in HeLa cells as presented in Figure 2D (only about 20% repaired in 6 hours). The authors should comment on this apparent discrepancy.

9. lines 174-177 and Figure 3A. Figure 2F shows that 2 hours after 10 J/m² UV, most 64PPs are indeed repaired. However, in this figure, a dose of 100 J/m² is used. How can the authors know that most 64PPs are repaired with this dose after 2 hours? Also, in line 177, the authors mention XPC depletion, but Figure 3A shows XPA depletion. Figure 3a is also mislabeled as Figure 3b. This should be corrected.

10. Figure 4B. It is unclear if the authors quantified the total of XPC signal or only the non-ubiquitinated XPC. On the blot, the non-ubiquitinated XPC band clearly decreases after UV, with a concomitant increase of the ubiquitinated forms.

11. Figure 5A and B. The quantification of the fluorescent intensity is unclear. This should be a ratio, i.e. intensity in the damage spot divided by nuclear background (as also indicated in the main text and in the methods) but the graphs show a percentage of XPC accumulation on the Y-axis. This should be clarified.

12. Figure 6A. It is striking that the authors do not observe any H3K4me3 in their H3 IP. With

H3K4me3 IP, H3 is co-precipitated but with H3 IP, H3K4me3 is not co-precipitated? Is this because the H3 antibody does not recognize the (tri-)methylated form of H3? According to Supplementary Figure 6, however, it does. It seems therefore also not correct to write that this antibody recognizes non-methylated H3. Please comment and correct.

13. In addition, in Figure 6B the authors show only a very marginal UV-induced increase of DDB2 co-IPing with H3K4me3 IPs, whereas from the blot presented in Figure 6A, no DDB2 could be detected without UV and a clear signal of DDB2 is observed after UV. The authors should clarify this.

14. Figure 6F. In the input lanes H3K4me3 is visible (most right lane), whereas the legends above the lanes indicate that no H3K4me3 was added.

15. Supplementary Figure 6A and B. Although these experiments appear to support the idea that XPC interacts with H3, they do not support that XPC preferentially interacts with H3K4me3 over H3. How did the authors quantify the XPC levels in figure B? Did they normalize to the H3 signal as they do in all other figures? Judging from figure A, this would suggest that XPC, in this particular experiment, has a preference for H3 and not H3K4me3.

16. Have the authors tried whether ASH1L or H3K4me3 visibly accumulates in a local UV damage spot in immunofluorescence?

Reviewer #2 (Remarks to the Author):

The research reported in this manuscript convincingly describes an unanticipated link between the UV-induced trimethylation of lysine 4 on histone H3 and the hand-off between the DDB2 and XPC; the GG-NER DNA damage recognition factors during repair of UV damage.

The authors show that DDB2 recruits the histone methyltransferase, ASH1L, to nucleosomes in chromatin resulting in the trimethylation of histone H3 in response to UV. This in turn is shown to recruit XPC to nucleosomal DNA, and this series of events results in the exchange between these two damage recognition factors at sites of DNA damage and that this exchange stimulates the activation of lesion removal by GG-NER. These steps drive the subsequent repair events during the removal of UV-induced CPDs from chromatin.

This is an important advance in our understanding of DNA damage recognition and repair of lesions in chromatin in human cells by NER, a major DNA repair pathway. The experiments conducted have been clearly described and the conclusions drawn are in general well justified.

I believe that following revision of the manuscript, taking into account the comments below, this report would be suitable for publication in Nature Communications

I have a number of suggestions that I trust might improve the report, making it more appealing and easier to digest for a broader readership beyond the immediate field.

Title: The title in my view is too generic and doesn't do justice to the findings reported. I suggest the something along the following lines: UV-induced histone trimethylation regulates the hand-off between DNA damage recognition factors during GG-NER in chromatin.

Abstract: Should also include the central hypothesis of the paper ie that histone trimethylation regulates the hand-off between GG-NER recognition factors during repair of lesions in chromatin.

Introduction: Is well written, but should also include the central hypothesis mentioned above. I

suggest moving the first paragraph from the Results section, which describes the rationale for examining histone methylation, to the end of the Introduction. This enables an easier transition from the rationale for examining histone methylation, to the exchange of damage recognition factors via the provision of docking sites following UV-induced histone methylation and the relaxation of chromatin.

Results: I suggest breaking up Figure 1 into two parts: a-d becomes Figure 1 and e-g becomes Figure 2. I would also suggest moving the screen for methyltransferases for UV sensitivity to Figure 1 (instead of Supplementary Info. It should also include the KD levels for proteins), and also show the effect on UV sensitivity, for ASH1L (currently shown in Figure 2d).

The results in Figure 1 demonstrate that DDB2 recruits ASH1L to chromatin (incidentally, to the subtitles in the text referring to Figure 1, the words 'to chromatin' should be added). However, how DDB2 attracts ASH1L to the chromatin in response to UV is not currently determined or discussed in any detail. Later experiments demonstrate that it is not the E3 ligase function of the UV-DDB complex that is responsible for this recruitment. Figure 1g describes an IP experiment on the chromatin-free fraction to show that DDB2 IP results in ASH1L co-IP, suggesting their interaction in the absence of UV damage to cells. It wasn't clear to me what the purpose of this experiment was (beyond the fact that depleting DDB2 results in the loss of ASH1L co-IP??). There could be many reasons for DDB2/ASH1L co-IP in the soluble fraction. However, I would like to suggest that an additional experiment that examines co-IP (by ChIP-western) not only on the soluble fraction, but also the chromatin fraction, both in the absence and following UV-induced DNA damage. I believe that this experiment could provide important insight into the nature of the interaction of DDB2 and ASH1L in chromatin both before and after UV damage. Observing changes in the co-localisation of these two proteins in chromatin could be informative. Can the authors please consider this suggestion? If there is some technical reason that makes this difficult then alternatively, I suggest moving Figure 1g to the supplementary info section.

Lines 229-231 of the Results it is stated that 'ASH1L recruitment is required for efficient recruitment to lesions and activation of XPC by ubiquitination, which in turn is necessary for the engagement of the XPD helicase.....' Whilst this is implied, it is not shown that XPC is activated by ubiquitination. Perhaps this can be speculated upon in the Discussion.

Finally, if it is possible, I would recommend moving the suggested model into the main text, as it will facilitate rapid understanding of the experiments and their significance.

Reviewer #3 (Remarks to the Author):

Review of Pogliano et al : ASH1 HMT promotes GG-NER

Summary

There are some potentially interesting observations here, mainly because ASH1 has not previously been implicated in cellular the response to UV light. The major problem is that ASH1 is known to be a transcriptional regulator, and so some of the differences observed here, in particular on UV survival and repair could be indirect. In addition I would be more cautious in the interpretation of small apparent differences in experiments that are by their nature non-linear and only semi-quantitative at best (such as western blots and immunofluorescent detection). I need further convincing that the data is robust before I would be able to recommend this paper as suitable for publication.

Specific points

The western blots for ASH1 are often cropped very tight (e.g. figure 1, and 6). Please provide full images of uncropped blots and evidence that the band that you are labelling as ASH1 is the true ASH1 band. How have these western blots been quantified? Quantification of western blots is problematic, as there are so many stages at which variability can be introduced. I personally don't

consider it likely that a 1.2x change (such as in fig1 c) can be reliably determined by western blotting, as in my lab the baseline error due to technical variation from blot to blot is greater than that, and we use a LiCor, which eliminates some of the nonlinearity (but in no way all the variability) inherent in western detection. How were these blots developed?

The conclusion that ASH1 chromatin recruitment is DDB2 dependent is weak, given that there is clearly less histone H3 control in the pertinent lane (fig1e).

Much of the data in Figure 2 and 3 could be explained if the transcriptional activity of ASH1 was required for efficient NER, can the authors comment?

I am surprised that only 20% cells are still CPD positive 6 hours after 100J/m² local irradiation (figure 2g). Was this assigned by eye or using an image analysis software? The latter would be preferable as less prone to operator bias. Only one cell is shown in each case, I would like to see a full field of view here to get a feel for levels of background and cell – cell variation.

What software are the authors using for UDS analysis? How are they setting thresholds for S phase and EdU background subtraction. It is unusual to do this with local irradiation.

As in figure 1, in figure 4 I am just not convinced that the western blot differences, particularly for XPC (and given the discrepancy with fig5), are sufficiently robust as to be interpretable. If the XPD can get to the damage at 1 hour (figure 5b) and the lesions persist in the ASH1 depletions, figure 2, then why would there be less XPD at the lesions later in the ASH1 depletion?

Again in figure 6 and 7 these differences in the westerns are very slight, and there are often changes in the control lanes (inputs) that make me dubious about the strengths of the conclusions drawn.

We wish to thank all three reviewers for their critical but constructive comments by which we were able to substantially increase the quality of our manuscript. Among other improvements, the reviewer's comments led to the following completely new experiments and data displays in the revised manuscript:

- New Fig. 1h: ChIP assay demonstrating the DDB2-ASH1L interaction in chromatin
- New Fig. 3b: Extended incubation period for the recovery of RNA synthesis confirming that ASH1L is not involved in TC-NER
- New Fig. 6a and 6b: Protein dynamics assay in chromatin showing that ASH1L is required for a stable association of XPC protein with CPD sites
- New Fig. 8c: Excision assay demonstrating that the histone-binding domain of XPC protein is critical for efficient CPD repair
- New Supplementary Fig. S4 confirming that ASH1L is not needed for the repair of 6-4PPs

Our point-by-point response to the referees' comments is detailed below.

Reviewer No. 1

Major issues:

1. Histone methyltransferase screen: To respond to this reviewer's comment, we added a sentence in the revised results section to make clear that our initial screen indeed suggested that several histone methyltransferases contribute to survival after UV exposure (p. 3, L. 91-92). In the discussion, we added the following explanation: "There are several potential mechanisms by which histone methyltransferases may participate in the cellular UV response. For example, the methylation of histone H3 at position K79 by DOT1L (for *Disruptor Of Telomeric silencing 1-Like*) has been shown to support transcription upon genotoxic stress without influencing DNA repair activity⁶². By contrast, we describe the ASH1L histone methyltransferase as an auxiliary factor that, like DDB2 (the product of the *XPE* gene), is indispensable for excision of CPD lesions." (p. 11, L. 349-354).

2. Dependence on DDB2: In the revised results section we added the following explanation to accommodate the results of CUL4A depletion: "Depletion of the CUL4A ubiquitin ligase scaffold affected partially the amount of DDB2 bound to chromatin, but without interfering with the extra UV-dependent chromatin localization of ASH1L. These findings indicate that in the absence of CUL4A ubiquitin ligase activity lower amounts of DDB2 are sufficient for the recruitment of ASH1L and, in any case, imply that the DDB2 subunit itself, rather than the associated ubiquitin ligase complex, is required for the UV-dependent ASH1L redistribution to chromatin." (p. 4, L. 123-129).

3. TC-NER assay: To address this reviewer's criticism, the TC-NER assay has been repeated with a longer incubation time of 6 h. Importantly, this assay based on EU incorporation is validated by CSB down regulation (conferring a TC-NER deficiency), which causes a nearly 80% reduction of RNA synthesis compared to the TC-NER-proficient control (new Fig. 3b). In view of this robust control, the revised assay clearly shows that ASH1L is not involved in the TC-NER pathway. We took care to indicate in each figure legend what cells were used (U2OS in Fig. 3a and 3b).

4. Chromatin binding of XPC protein: The criticism regarding the former Fig. 4d (Fig. 6d in the revised manuscript) probably originated from the reviewer's believe that our quantifications did not take into account the overall higher XPC level in ASH1L-depleted cells. In response, we improved the revised text as follows: "Cells were collected 1 and 3 h after treatment and, without prior fixation, subjected to 0.3-M NaCl extraction to remove free proteins that are not or only loosely bound to chromatin. The remaining chromatin was

analyzed by gel electrophoresis and immunoblotting, and quantifications took into account the generally higher level of XPC protein in ASH1L-depleted cells. As reported before⁵⁸, a substantial proportion of the XPC subunit is constitutively bound to chromatin both in control cells and ASH1L-depleted cells even without radiation. In control cells, the level of XPC protein recruited to chromatin is increased 1 h after UV exposure relative to the unchallenged situation (Fig. 6c, 6d). However, in ASH1L-depleted cells, this extra UV-dependent XPC recruitment to chromatin compared to the respective unirradiated reference is much lower, resulting in higher proportions of XPC in the fraction of free proteins" (p. 8, L. 258-268). Importantly, in the revised manuscript the outcome of this biochemical assay is confirmed by the new in situ immunofluorescence experiment of Fig. 6b, both demonstrating that ASH1L is needed for a stable interaction of XPC protein with CPD sites.

5. Protein dynamics: The apparent contradiction spotted by the reviewer is addressed by the new Fig. 6b (following the method depicted in the new Fig. 6a), demonstrating that ASH1L is needed for the stable association of XPC protein with CPD sites. Instead of carrying out FRAP assays with fluorescent protein constructs, we preferred to test the nuclear dynamics of endogenous XPC protein as outlined in Fig. 6a.

6. Functional test: The functional significance of the D748K mutation has been tested and the results (new Fig. 8c) demonstrate that this charge inversion in the beta-turn motif confers a strong CPD excision defect.

Minor issues:

7. The blot showing H3K4 methylation is now presented with 1- and 3-h time points, as requested by the reviewer. Also, a separate blot in Fig. 2a shows that the UV-induced H3K4 methylation depends on DDB2 (as expected from the findings of Fig. 1).

8. As indicated in the revised text, we believe that the readout of Fig. 2G (CPD excision from UV lesion spots) may be influenced by the lower signal-to-noise ratio of this method compared to the enzyme immunoassay (p. 6, L. 170-172). Nevertheless, this assay nicely confirms the CPD excision defect of ASH1L-depleted cells.

9. The new Supplementary Fig. S4 shows the rapid kinetics of 6-4PP excision from UV lesion spots generated by irradiating through filters with a dose of 100 J/m². The incorrect labeling of Fig. 3a has been amended and the error in the text describing this figure (mentioning XPC instead of XPA) has been corrected.

10. We make clear in the legend to Fig. 4b that the quantification includes both the ubiquitinated and the non-ubiquitinated forms of XPC protein (p. 25, L. 860).

11. In the revised legend to Fig. 5, we now clearly describe the quantification procedure. This explanation reads as follows: "Quantifications show the percentage of CPD-positive cells containing visible protein spots, and ratios of fluorescence intensity of each protein spot divided by the nuclear background (n=3 with at least 100 cells in each experiment). The intensity ratio seen with siNC controls is set to 100%" (p. 25, L. 866-869).

12. We corrected the description of the experiments of Fig. 7a (formerly Fig. 6a) to acknowledge that the generic antibody against histone H3 may also precipitate modified forms of the histone (p. 9, L. 289-292).

13. The quantification of DDB2 levels in Fig. 7b (formerly Fig. 6b) was incorrect and we apologize for this inadvertent mistake. The quantifications have now been repeated and they reflect the expected poor binding of DDB2 to undamaged chromatin.

14. The former Fig. 6f, now Fig. 7f has been replaced with a new experiment. The input fractions contain only a low level of endogenous H3K4me3 present in the sf9 whole-cell lysate.

15. The former Supplementary Fig. S6 has been removed.

16. Unfortunately, so far we have not been able to identify the accumulation of ASH1L or H3K4me3 in UV lesion spots. We think that existing protocols need to be optimized or perhaps the available antibodies are not suited for this purpose.

Reviewer No. 2

Major issues:

Title: The title has been revised following the reviewer's recommendation, but in a way to stay within the limits of 15 words.

Abstract: The abstract has been revised following the reviewer's recommendation (see p. 1, L. 29-30) but taking care to stay within the 150-word limit.

Introduction: The introduction has been revised by inclusion of the first paragraph of the results section as recommended by the reviewer (p. 3, L. 74-79).

Results: We tried to break up Fig. 1 but were not satisfied with the outcome. For example, Fig. 1b and 1f both show the chromatin recruitment of ASH1L and these panels should stay within the same figure. Supplementary Fig. S1 is only an initial siRNA screen without corresponding protein levels (except for ASH1L).

Figure 1: The chromatin-immunoprecipitation assay (ChIP) requested by this reviewer has been carried out and is now shown as new Fig. 1h. This ChIP assay demonstrates the expected interaction between DDB2 and ASH1L in the chromatin context.

Lines 229-231: The criticized statement regarding XPC ubiquitination has been deleted in the revised manuscript. A careful re-evaluation of our findings indicates that the ubiquitination reaction is not influenced by ASH1L.

Model: In the revised manuscript, the model has been moved to the main text and presented as Fig. 9, as suggested by the reviewer.

Reviewer No. 3

Specific points:

– Western blots: Besides providing full images of all blots in Supplementary Fig. S7 and S8, the reason for the tight cropping of some blots showing ASH1L is demonstrated in the new Supplementary Fig. S3f. Briefly, the immunoblots analyzing chromatin-associated ASH1L contain a cross-reacting band that migrates only slightly faster than ASH1L protein. In Fig. S3f, depletion of ASH1L by siRNA highlights this cross-reacting band marked by asterisks. This problem only relates to ASH1L in chromatin fractions and, as shown in Fig. S3c, does not extend to the detection of ASH1L protein in whole-cell lysates. The legend to the new Supplementary Fig. S3f reads as follows "Non-specific band, indicated by the asterisks, appearing when immunoblotting the chromatin fraction against ASH1L. The chromatin of HeLa cells transfected with siASH1L (or non-coding siNC) was immunoblotted to

demonstrate that only the upper of two closely migrating bands represents the ASH1L protein". The revised methods section outlines how the western blots were prepared, developed and quantified (p. 14, L. 449-458).

– Quantifications: In response to the reviewer's skepticism regarding the Western quantifications, we added below the blots of Fig. 1c and 1e the measured protein levels normalized to histone H3 used as the internal standard. These findings were reproduced in 4-6 independent experiments and the differences are statistically significant (Fig. 1d and 1f). Also, the new Fig. 1h shows by ChIP that there is indeed an interaction between DDB2 and ASH1L that is responsible for the recruitment of ASH1L to the chromatin of UV-damaged cells.

– Transcription: The revised manuscript explains in more detail the reasons why we conclude that ASH1L is not involved in the TC-NER pathway. First, the drop of CPD excision upon ASH1L depletion is too strong to be explained with a TC-NER defect (p. 5, L. 157-160). Second, we provide direct evidence, controlled by a CSB knockdown, that ASH1L is not involved in TC-NER activity (new Fig. 3b). Third, the ASH1L depletion does not reduce the expression of GG-NER factors (see Fig. 4).

– Immunofluorescence analysis of CPD repair: The revised Supplementary now contains wide-field views of UV lesion spots after different repair periods (new Supplementary Fig. S6). These images show that, at the 6-h time point, the CPD repair defect of ASH1L-depleted cells can be easily and immediately seen by eye. In any case, the revised manuscript also describes in detail how immunofluorescence signals, including those generated in UDS assays, were analyzed (p. 14, L. 441-447 and p. 16, L. 513-519).

– XPD Recruitment: We believe that the new Fig. 6b (using the method depicted in the new Fig. 6a) indicates a clear mechanism of action of ASH1L in that this histone methyltransferase is needed for a stable interaction of XPC protein with CPD sites in the chromatin context. In the absence of ASH1L, XPD can still be recruited to 6-4PPs at the 1-h time point after UV irradiation. However, at 3 h after irradiation the 6-4PPs are nearly completely repaired but ASH1L is needed for the repair of CPDs. This mechanism explains why the XPD recruitment is more strongly affected by the ASH1L depletion at 3 h than at 1 h after irradiation (p. 9, L. 273-276 and model of Fig. 9).

– It is difficult to handle a generic criticism like "...in figure 6 and 7 these differences in the westerns are very slight, and there are often changes in the control lanes (inputs) that make me dubious about the strengths of the conclusions drawn". In Fig. 7 and 8 of the revised manuscript (formerly Fig. 6 and 7), the detected differences between conditions are actually substantial (see for example Fig. 7a and 8e) and the conclusion is confirmed by a functional excision assay (new Fig. 8c) clearly demonstrating that the histone-interacting motif of XPC determines the efficiency of CPD excision.

REVIEWERS' COMMENTS:

Reviewer #1 (Remarks to the Author):

In this revised manuscript, the authors have carried out additional experiments that further corroborate their finding that the methyltransferase ASH1L and H3K4 methylation promote processing of CPD lesions via DDB2 and stable association of XPC in Nucleotide Excision Repair. Most of our concerns were addressed in a satisfactory fashion.

One remaining minor concern regards experiments shown in Fig 1g and h. From these figures, it seems clear that DDB2 and ASHL1 interact, but there is no evidence of a 'damage-dependent' interaction. Already without damage there is an interaction (figure 1g) and the stronger ASHL1 band observed after UV can simply be caused by the fact that more DDB2 is present in chromatin after UV (figure 1h). This should be adequately discussed in the text in lines 136-139.

Reviewer #2 (Remarks to the Author):

The authors have adequately addressed my comments and suggestions. In particular, they conducted a key experiment that significantly enhances their evidence for DDB2 recruitment of ASH1L to UV damaged chromatin. I am still of the opinion that this result could have been more prominently reported. It is somewhat buried in the details as Figure 1H. It might also have been referred to in the authors' reply to the other reviewers' comments, particularly reviewer 3, as it significantly addresses certain of their concerns. I am satisfied that the revised manuscript has been markedly improved following the extensive revision.

Reviewer #3 (Remarks to the Author):

The authors have made a significant effort in improving this manuscript. The new experiment presented in Figure 6 is well thought out. Although I am still of the view that some of the western blot data is representing only small effects, and is thus open to over-interpretation, the large excision defects, cell survival effects and alterations to protein recruitment in situ caused by the ASH1L depletions cannot be denied. As such this paper is worthy of publication as it will be of interest to anyone interested in cellular UV responses.

We wish to thank all three reviewers for their critical and constructive comments by which we were able to substantially increase the quality of our manuscript.

Our point-by-point response to the referees' remaining comments is detailed below.

Reviewer No. 1

The reviewer's criticism of our description of the IP and ChIP experiments of Fig. 1g and 1h is correct in that the interaction between DDB2 and ASH1L is not UV-dependent. However, the recruitment of DDB2 to chromatin is damage-dependent. Therefore, the relevant conclusion in the main text has been corrected as follows: "Thus, an interaction occurs between these two factors such that DDB2 is able to mediate the relocation of ASH1L to UV lesions in the chromatin context" (p. 5, L. 5-8 of the revised manuscript with track changes feature).

Reviewer No. 2

This reviewer did not like the fact that the IP and ChIP experiments are "buried" in the details of Fig. 1. To emphasize more prominently the importance of these protein interaction experiments, we include an additional reference to Fig. 1g (IP) and Fig. 1h (ChIP) in the Discussion section (p. 12 of the revised manuscript with track changes feature).

Reviewer No. 3

This reviewer was satisfied with the revisions and did not request further changes.